# Enabling Optimal Decisions in Rehearsal Learning under CARE Condition

**Wen-Bo Du** [1 2]  **Hao-Yi Lei** [1 2]  **Lue Tao** [1 2]  **Tian-Zuo Wang** [1 2]  **Zhi-Hua Zhou** [1 2]

## Abstract

In the field of machine learning (ML), an essential type of decision-related problem is known as AUF (Avoiding Undesired Future): if an ML model predicts an undesired outcome, how can decisions be made to prevent it? Recently, a novel framework called *rehearsal learning* has been proposed to address the AUF problem. Despite its utility in modeling uncertainty for decision-making, it remains unclear *under what conditions* and *how* optimal actions that maximize the *AUF probability* can be identified. In this paper, we propose *CARE* (CAnonical REctangle), a condition under which the maximum AUF probability can be achieved. Under the CARE condition, we present a projection-Newton algorithm to select actions and prove that the algorithm achieves superlinear convergence to the optimal one. Besides, we provide a generalization method for adopting the algorithm to AUF scenarios beyond the CARE condition. Finally, we demonstrate that a closed-form solution exists when the outcome is a singleton variable, substantially reducing the time complexity of decision-making. Experiments validate the effectiveness and efficiency of our method.

## 1. Introduction

Machine Learning (ML) models have achieved great success in various real-world prediction tasks (LeCun et al., 2015). Instead of solely focusing on the prediction, Zhou (2022) emphasizes another important issue, *i.e.*, if the prediction of an ML model is undesired, how to find effective actions to prevent it from happening. This problem is known as the *AUF (avoiding undesired future)* problem (Zhou, 2022). Consider an autonomous drone delivery system scenario as an example. Imagine that the system has trained an ML

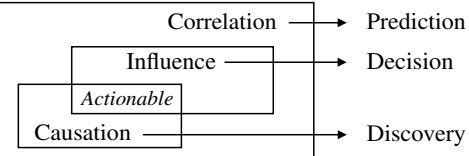

*Figure 1.* Relations between correlation, influence, and causation (reproduced from Zhou (2022)).

model that takes real-time environmental data (denoted as $\mathbf{X}$) as input and predicts the risk of package loss (denoted as $\mathbf{Y}$). For a particular delivery, if the prediction indicates a high risk of loss, the system will consider altering values of some intermediate control variables (denoted as $\mathbf{Z}$), such as altitude, speed, or route, to mitigate the risk. Since experimenting with real deliveries to explore better actions is costly and potentially results in lost packages, different alterations on $\mathbf{Z}$ must be carefully considered. Let $\mathcal{S}$ denote the desired region of $\mathbf{Y}$ where the risk of package loss is sufficiently low; the AUF problem can be interpreted as finding the optimal alteration $\mathbf{z}^{\xi}$ that maximizes the probability of $\mathbf{Y} \in \mathcal{S}$, which is called the *AUF probability* ($\mathbb{P}_{\mathrm{AUF}}$).

Generally, decision-makers lack access to an explicit formula for $\mathbb{P}_{\mathrm{AUF}}$ *w.r.t.* alterations, which complicates the selection of effective alterations. To model $\mathbb{P}_{\mathrm{AUF}}$, we need to understand the relations among variables $\{\mathbf{X}, \mathbf{Z}, \mathbf{Y}\}$. While correlation is often sufficient for prediction in ML models, effective decision-making demands a more nuanced understanding of variable relations. In certain decision-making contexts, causal relations are helpful to model these relations (Bareinboim et al., 2015; Lattimore et al., 2016). In practice, identifying causal relations is essentially challenging, as it typically relies on untestable assumptions (Spirtes et al., 2000; Schölkopf et al., 2021). However, a complete causal understanding is not a prerequisite for effective decision-making, as evidenced by human decision-makers who could successfully make decisions without such comprehensive knowledge (Zhou, 2022). Furthermore, even when causal factors are identified, they may not be actionable for decision-making purposes. Additionally, variables that influence outcomes are not necessarily causal factors. Recognizing that *correlation* is insufficient and *causation* is somewhat luxurious for decision-making, Zhou (2023) propose the *influence relation* as an intermediate concept between correlation and causation, as depicted in Fig. 1.

[1]National Key Laboratory for Novel Software Technology, Nanjing University, China [2]School of Artificial Intelligence, Nanjing University, China. Correspondence to: Zhi-Hua Zhou <zhouzh@lamda.nju.edu.cn>.

*Proceedings of the 42nd International Conference on Machine Learning*, Vancouver, Canada. PMLR 267, 2025. Copyright 2025 by the author(s).

*Table 1.* Time-complexity for decision-making, $|\mathbf{Y}| = 1$

| ALG. | QIN ET AL., 2023 | DU ET AL., 2024 | OURS |
|---|---|---|---|
| TIME | $\notin \mathcal{O}\left(p^k\right), \forall k \in \mathbb{N}$ | $\mathcal{O}\left(p^3\right)$ | $\mathcal{O}\left(p\right)$ |

Building on this, Qin et al. (2023) develop the *rehearsal learning* framework to learn influence relations and provide a corresponding decision-making algorithm based on the relations, and Du et al. (2024) consider the decision cost and enhance the computational efficiency. While the learned relations aid in selecting effective alterations, directly optimizing $\mathbb{P}_{\mathrm{AUF}}$ remains challenging due to the lack of an analyzable functional form in most scenarios. Consequently, existing rehearsal learning methods typically identify plausible alterations by ensuring $\mathbb{P}_{\mathrm{AUF}}$ exceeds a predetermined threshold $\tau$, formulated as $\mathbb{P}_{\mathrm{AUF}} \geq \tau$.

The existing rehearsal learning approaches suffer from two critical shortcomings. First, the threshold-based strategy for maximizing the AUF probability is unsuitable. Let $\mathrm{C}_\tau$ represent the constraint $\mathbb{P}_{\mathrm{AUF}} \geq \tau$ and $\mathbb{P}_{\mathrm{A}}^\star$ denote the maximum achievable AUF probability through alterations. When $\tau > \mathbb{P}_{\mathrm{A}}^\star$, the feasible domain of $\mathrm{C}_\tau$ becomes empty, causing existing rehearsal learning approaches fail to find feasible alterations. Conversely, when $\tau < \mathbb{P}_{\mathrm{A}}^\star$, solutions satisfying $\mathrm{C}_\tau$ may be suboptimal, as they are not guaranteed to maximize the AUF probability. Second, the computational burden of constructing the feasible region satisfying $\mathrm{C}_\tau$ is prohibitive, especially for high-dimensional problems. Specifically, when constructing the feasible domain of $\mathrm{C}_\tau$, the time complexity of existing sampling-based approach cannot be bounded by any polynomial function of $p$, and the method in Du et al. (2024) exhibits cubic complexity $\mathcal{O}(p^3)$ (where $p$ represents the dimension of $\mathbf{V}$) in cases where $|\mathbf{Y}| = 1$, as illustrated in Tab. 1.

To address the aforementioned shortcomings, we propose a novel approach to directly maximize the AUF probability $\mathbb{P}_{\mathrm{AUF}}$. Specifically, we establish the *CARE* (CAnonical REctangle) condition for the AUF problem, a condition that is practical in real-world tasks. Under the CARE condition, we show that $\mathbb{P}_{\mathrm{AUF}}$ can be explicitly formulated as a function of the selected alteration $\mathbf{z}^\xi$ given observed variables $\mathbf{x}$ and influence relations parameterized by $\boldsymbol{\theta}$, which eliminates the need to construct the feasible domain satisfying $\mathrm{C}_\tau$ and facilitates direct optimization. Although this function is not inherently concave, we prove that applying a $\log$ transformation ensures concavity, thereby guaranteeing the existence of a global maximum of $\mathbb{P}_{\mathrm{AUF}}$. While the transformed function remains complex, both the gradient vector and Hessian matrix are always computable. Leveraging these properties, we develop a projection-Newton method to select decision alterations for the AUF problem, which is proven to achieve superlinear convergence to the optimal $\mathbf{z}^\xi$

that maximizes $\mathbb{P}_{\mathrm{AUF}}$. For cases where the CARE condition does not hold, we introduce an inner embedding technique to construct a concave lower bound of $\log \mathbb{P}_{\mathrm{AUF}}$, which satisfies the CARE condition and can be used as a surrogate for optimization. Last but not least, we demonstrate that the CARE condition always holds when the desired AUF region is an interval and constructively derive a closed-form solution for the optimal $\mathbf{z}^\xi$ when $|\mathbf{Y}| = 1$, significantly reducing the time complexity, as illustrated in Tab. 1. Finally, experimental results validate the effectiveness and efficiency of our approach on both synthetic and real-world datasets.

Our contributions can be summarized as follows:

1. We propose the CARE condition for AUF problem, under which the logged AUF probability can be proven to be a concave function *w.r.t.* the selected alteration $\mathbf{z}^\xi$.

2. We present a projection-Newton algorithm to select the optimal alteration, and we prove that the algorithm can achieve a superlinear convergence rate.

3. We provide an inner embedding methodology to handle the AUF problem not satisfying the CARE condition. Experiments validate the effectiveness of our methods.

4. We prove the existence of a closed-form solution when $|\mathbf{Y}| = 1$, in which case the time complexity can be significantly reduced for addressing the AUF problem.

## 2. Preliminaries and problem setup

A probabilistic graphical model called structural rehearsal model (SRM) is proposed by Qin et al. (2023) to characterize influence relations among variables in the AUF problem. Influence relations can capture mutual dependencies between variables while accommodating dynamic environments, making it more suitable for decision-making. The SRM comprises a set of rehearsal graphs and associated parameters $\{\langle G_t, \boldsymbol{\theta}_t \rangle\}$. Focusing on the optimal action in specific decision rounds, we consider $G_t \triangleq G$ in this work, which can be straightforwardly generalized to multi-round scenarios. Detailed definition of SRM is listed in Appx. A.1.

The rehearsal graph $G$ models the qualitative influence relations among variables, which is denoted by $G = (\mathbf{V}, \mathbf{E})$. Specifically, $\mathbf{V}$ represents the variable set of the AUF problem and $\mathbf{E}$ represents the edges characterizing influence relations among variables. There are two types of edges in $G$, a directional edge $X \to Y$ means that $X$ influences $Y$, and a bi-directional edge $X \leftrightarrow Y$ means that $X$ and $Y$ are mutually influenced. For example, sunlight unilaterally influences the plant growth, whereas rainfall and river flow are mutually influenced, as changes in either one affect the other. Besides, the corresponding structural equations of variable $V_j$s can be parameterized by $\{\boldsymbol{\beta}_j, \sigma_j^2\}_{j=1}^{|\mathbf{V}|} \subseteq \boldsymbol{\theta}$:

$$V_j := f_j\left(\mathrm{PA}_j^G; \boldsymbol{\beta}_j\right) + \varepsilon_j, \tag{1}$$

where $V_j \in \mathbf{V}$ denotes the $j$-th vertex of the graph $G$, $\mathrm{PA}_j^G \triangleq \{u \mid u \to V_j \text{ in } G\}$ represents parents of $V_j$, and the additive noise $\varepsilon_j$ follows $\mathcal{N}(0, \sigma_j{}^2)$. In this work, we focus on a basic yet essential class of the AUF problem, characterized by linear structural equations $f_j$s in Eq. (1), *i.e.*, $V_j := \boldsymbol{\beta}_j^T \mathrm{PA}_j + \varepsilon_j$, where $\boldsymbol{\beta}_j \in \mathbb{R}^{|\mathrm{PA}_j|\times 1}$, consistent with prior studies (Qin et al., 2023; Du et al., 2024).

Besides, the decision-making process centers on identifying appropriate *alterations*, which are decision actions specified by human decision-makers as sets of vertex-value pairs. For instance, as shown in Fig. 6 (provided in Appx. A.1), $\xi \leftarrow \{Z_1 = z_1\}$ in Fig. 6(b) demonstrates such an alteration. Meanwhile, the execution of an alteration is formalized through a *rehearsal operation*, denoted as $Rh(\cdot)$, which modifies the original graph structure as shown in Fig. 6(b)-Fig. 6(d). The rehearsal operation removes all incoming influence links that point into any vertices contained in $\xi$, and fixes the values in $\xi$ to their associated vertices; while this operation maintains the influence relations among all other vertices in the resulting graph $G^{Rh(\xi)}$.

For the AUF problem, the goal is to find an optimal alteration that can maximize the AUF probability after performing it. As the generation of variables is parameterized by $\boldsymbol{\theta}$ in an SRM, and the observation of $\mathbf{x}$ happens before making decisions, the AUF probability is conditional on $\boldsymbol{\theta}, \mathbf{x}$. Also, in practice, the alteration range is limited for the variables $\mathbf{z}$. Thus, the AUF problem can be formulated as the following constrained optimization problem:

$$\underset{\mathbf{z}^\xi}{\arg\max} \quad \mathbb{P}\left(\mathbf{Y} \in \mathcal{S} \mid \mathbf{x}, \boldsymbol{\theta}, Rh(\mathbf{z}^\xi)\right) \\ \text{s.t.} \quad \mathbf{z}_{\text{left}}^\xi \leq \mathbf{z}^\xi \leq \mathbf{z}_{\text{right}}^\xi. \tag{2}$$

The constraint in Eq. (2) uses a compact element-wise notation, where $\mathbf{z}_{\text{left}}^\xi \leq \mathbf{z}^\xi \leq \mathbf{z}_{\text{right}}^\xi$ means that each element of $\mathbf{z}^\xi$ must satisfy its corresponding lower and upper bounds specified in $\mathbf{z}_{\text{left}}^\xi$ and $\mathbf{z}_{\text{right}}^\xi$ respectively.

# 3. Our approach

In this section, we propose our approach to identify decision actions that maximize $\mathbb{P}_{\text{AUF}}$. The primary challenge lies in the lack of an analyzable functional form and the potential irregularity of $\mathbb{P}_{\text{AUF}}$, making direct optimization intractable. To address this, we propose a special type of desired region $\mathcal{S}$, under which we prove that $\mathbb{P}_{\text{AUF}}$ admits an analyzable functional form and exhibits convexity after applying a negative log transformation, enabling efficient decision optimization. For AUF cases with irregular $\mathcal{S}$, we present a generalization method to optimize an upper bound of the negative log AUF probability as a surrogate, thereby extending our approach to broader application scenarios.

In what follows, we establish the key condition, *i.e.*, the

CARE condition, and prove that $\mathbb{P}_{\text{AUF}}$ under this condition can be explicitly formulated and exhibits convexity *w.r.t.* the decision alterations after a negative log transformation.

## 3.1. CARE condition for AUF problem

In this subsection, we introduce the CAnonical REctangle (CARE) condition for the desired region $\mathcal{S}$. The motivation of the condition stems from the observation that the functional form of AUF probability $\mathbb{P}\left(\mathbf{Y} \in \mathcal{S} \mid \mathbf{x}, \boldsymbol{\theta}, Rh(\mathbf{z}^\xi)\right)$ depends on the shape of the desired region $\mathcal{S}$. We start by defining the convex polytope in Def. 3.1, a superclass of the canonical rectangle in the CARE condition.

**Definition 3.1** (**Convex polytope**). The region $\mathcal{P}$ is a convex polytope in $\mathbb{R}^d$ iff it could be written as $\mathcal{P}(\mathbf{M}, \mathbf{d}) \triangleq \{\mathbf{s} \in \mathbb{R}^d : \mathbf{M}\mathbf{s} \leq \mathbf{d}\}$ with $\mathbf{M} \in \mathbb{R}^{n\times d}$ and $\mathbf{d} \in \mathbb{R}^{n\times 1}$.

In general, a convex polytope expresses a convex region that is formed by the intersection of multiple half-spaces divided by hyperplanes. In this sense, the empty set is contained in the definition and the convex polytope could be an open region. Based on this, we define the canonical rectangle in Def. 3.2, which is a subclass of convex polytope.

**Definition 3.2** (**Canonical rectangle**). Let $\mathcal{P}(\mathbf{M}, \mathbf{d})$ denote a convex polytope in $\mathbb{R}^d$, $\mathbf{G} \succ 0$ denote a positive definite matrix in $\mathbb{R}^{d\times d}$. $\mathcal{P}(\mathbf{M}, \mathbf{d})$ is called to be a *canonical rectangle w.r.t.* matrix $\mathbf{G}$ iff the following holds:

1. $\mathcal{P}(\mathbf{M}, \mathbf{d})$ expresses a non-empy closed region;
2. $\mathbf{M} \in \mathbb{R}^{2d\times d}$ with full rank, $\mathbf{d} \in \mathbb{R}^{2d\times 1}$; and
3. $\exists$ permutation matrix $\mathbf{P}_\sigma \in \mathbb{R}^{2d\times 2d}$ and diagonal matrix $\boldsymbol{\Lambda} \in \mathbb{R}^{2d\times 2d}$ such that $\mathbf{M} = \boldsymbol{\Lambda}\mathbf{P}_\sigma(\mathbf{I}, -\mathbf{I})^\mathsf{T}\mathbf{Q}$, where $\mathbf{G} = \mathbf{Q}^\mathsf{T}\mathbf{D}\mathbf{Q}$ is the eigen-decomposition of $\mathbf{G}$.

**Remark 3.3.** Note that if a convex polytope $\mathcal{P}(\mathbf{M}, \mathbf{d})$ is canonical *w.r.t.* $\mathbf{G} \in \mathbb{R}^{d\times d}$, then the polytope $\mathcal{P}(\mathbf{M}, \mathbf{d}) = \{\mathbf{s} \in \mathbb{R}^d : \mathbf{M}\mathbf{s} \leq \mathbf{d}\}$ expresses a (hyper-)rectangle region in $\mathbb{R}^d$. We provide two geometric intuitions of the canonical rectangle. First, edges of the rectangle are parallel to axes of the coordinate system induced by $\mathbf{Q}$, as shown in Fig. 2(a). Second, surfaces of the rectangle are parallel to axial planes of the ellipse $\mathbf{s}^\mathsf{T}\mathbf{G}\mathbf{s} = 1$, as shown in Fig. 2(b). Without loss of generality, for $\mathbf{G} = \mathbf{Q}^\mathsf{T}\mathbf{D}\mathbf{Q}$, we assign $\mathbf{P}_\sigma = \mathbf{I}$ and $\boldsymbol{\Lambda} = \text{diag}(\mathbf{D}^{-\frac{1}{2}}, \mathbf{D}^{-\frac{1}{2}})$ (thus $\mathbf{M} = (\mathbf{I}, -\mathbf{I})^\mathsf{T}\mathbf{D}^{-\frac{1}{2}}\mathbf{Q}$), since $\mathbf{d}$ can fit $\mathbf{M}$ by permutation and rescaling.

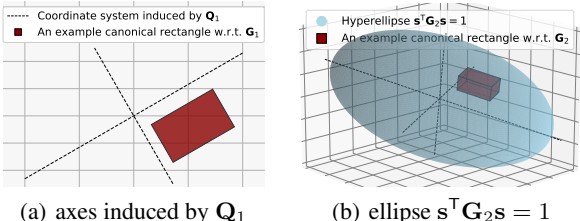

(a) axes induced by $\mathbf{Q}_1$      (b) ellipse $\mathbf{s}^\mathsf{T}\mathbf{G}_2\mathbf{s} = 1$

*Figure 2.* Geometric intuitions of the canonical rectangle

Recall from Eq. (2) that the goal is to find alterations maximizing the AUF probability. Since optimizing a probabilistic target is generally intractable, we seek conditions under which the AUF probability becomes analyzable. Building on the canonical rectangle, we propose the CARE condition in Def. 3.5 to achieve this goal. Before that, we present Lemma 3.4, which links the target variables $\mathbf{Y}$ to the observation $\mathbf{x}$, the alteration $\mathbf{z}^\xi$, and parameters $\boldsymbol{\theta}$.

**Lemma 3.4** (Qin et al., 2023). *Given observed $\mathbf{x}$ and parameter set $\boldsymbol{\theta}(\triangleq \{\{\boldsymbol{\beta}_j\}s, \boldsymbol{\Sigma}\})$ of the SRM, it holds that $\mathbf{Y} = \mathbf{A}\mathbf{x} + \mathbf{B}\mathbf{z}^\xi + \mathbf{C}\boldsymbol{\varepsilon}$, where $\mathbf{A}, \mathbf{B}, \mathbf{C}$ are constant matrices of appropriate shapes based on $\boldsymbol{\theta}$, $\mathbf{z}^\xi$ are intermediate variables with alteration $\xi$, and $\boldsymbol{\varepsilon} = [\varepsilon_1, \ldots, \varepsilon_{|\mathbf{V}|}] \sim \mathcal{N}(\mathbf{0}, \boldsymbol{\Sigma})$.*

**Definition 3.5** (CARE condition). The AUF problem satisfies the CARE (CAnonical REctangle) condition iff the associated desired region $\mathcal{S}$ is a canonical rectangle $\mathcal{S}(\mathbf{M}, \mathbf{d})$ *w.r.t.* matrix $\mathbf{C}\boldsymbol{\Sigma}\mathbf{C}^\mathsf{T}$ with $\mathbf{C}, \boldsymbol{\Sigma}$ defined in Lemma 3.4.

This condition is prevalent in real-world scenarios, particularly when the dimensions of $\mathbf{Y}$ are mutually independent. Considering a two-dimensional random variable $\mathbf{Y} = \{Y_1, Y_2\}$ as an example, if $Y_1 \perp\!\!\!\perp Y_2$ after alteration and the desired region $\mathcal{S} = \{a_1 \leq Y_1 \leq b_1, a_2 \leq Y_2 \leq b_2\}$, then the AUF problem satisfies the CARE condition. Under the CARE condition, the AUF probability can be formulated as an analyzable function, and its negative log form exhibits convexity, as shown in the following Thm. 3.6.

**Theorem 3.6.** *Under CARE condition that $\mathcal{S} \triangleq \mathcal{S}(\mathbf{M}, \mathbf{d})$, let $\mathbb{P}_A$ denote AUF probability $\mathbb{P}(\mathbf{Y} \in \mathcal{S} \mid \boldsymbol{\theta}, \mathbf{x}, Rh(\mathbf{z}^\xi))$, $\mathbf{k}_j^\mathsf{T}$ denote the $j$-th row of matrix $\mathbf{M}\mathbf{B}$, and $b_j$ denote the $j$-th component of vector $\mathbf{M}\mathbf{A}\mathbf{x} - \mathbf{d}$ ($\mathbf{A}, \mathbf{B}$ defined in Lemma 3.4). Then the following asserts hold:*

1. *Let $|\mathbf{Y}| = d$, the AUF probability can be expressed as:*
$$\mathbb{P}_A = \prod_{i=1}^{d} \left\{ \Phi\left(-\mathbf{k}_i^\mathsf{T}\mathbf{z}^\xi - b_i\right) - \Phi\left(\mathbf{k}_{i+d}^\mathsf{T}\mathbf{z}^\xi + b_{i+d}\right) \right\}.$$

2. *For any bounded alteration $\mathbf{z}^\xi$, $\mathbb{P}_A > 0$ and the function $\ell\left(\mathbf{z}^\xi\right) = -\log\left(\mathbb{P}_A\right)$ is convex w.r.t. $\mathbf{z}^\xi$.*

Thm. 3.6 provides the functional form of the AUF probability under the CARE condition defined in Def. 3.5. As a consequence, Cor. 3.7 reveals that the AUF problem can be transformed into a convex optimization.

**Corollary 3.7.** *Let $d$ denote $|\mathbf{Y}|$. Under the CARE condition, the AUF problem defined in Eq. (2) can be equivalently transformed into a convex optimization as follows:*

$$\arg\min_{\mathbf{z}^\xi} \quad \sum_{i=1}^{d} -\log\left\{\Delta_i(\mathbf{z}^\xi)\right\} \tag{3}$$
$$\text{s.t.} \quad \mathbf{z}_{\text{left}} \leq \mathbf{z}^\xi \leq \mathbf{z}_{\text{right}},$$

*where $\Delta_i(\mathbf{z}^\xi) = \Phi\left(-\mathbf{k}_i^\mathsf{T}\mathbf{z}^\xi - b_i\right) - \Phi\left(\mathbf{k}_{i+d}^\mathsf{T}\mathbf{z}^\xi + b_{i+d}\right)$, with $\mathbf{k}_j$ and $b_j$ defined as in Thm. 3.6.*

### 3.2. Projection-Newton for addressing the AUF problem

In this subsection, we present a projection-Newton method to solve the AUF problem with theoretical guarantees. The proof of the theoretical result is provided in Appx. C.

As illustrated in Cor. 3.7, the AUF problem can be transformed into a convex optimization under the CARE condition. To solve the AUF problem, there remain two challenges: (a) although Thm. 3.6 can express the AUF probability by known elements, the functional expression of the associated Gaussian CDF $\Phi(\cdot)$ remains complex; and (b) AUF scenarios exist beyond the CARE condition, in which case the transformation in Eq. (3) does not hold anymore. In what follows, we propose an algorithm to accurately solve the transformed problem in Eq. (3) and provide a generalization method of our algorithm for AUF scenarios that do not satisfy the CARE condition in the following subsection.

The formula of the gradient $\mathbf{g}$ and the Hessian $\mathbf{H}$ of the optimization target $\ell(\mathbf{z}^\xi) = \sum_{i=1}^{|\mathbf{Y}|} -\log\left\{\Delta_i(\mathbf{z}^\xi)\right\}$ is presented by Eq. (10) in Appx. C, which are associated with Gaussian CDF $\Phi(\cdot)$ and Gaussian PDF $\phi(\cdot)$. Although the functional expression of $\Phi(\cdot)$ is too complex to analyze, its exact value at any specific point can be accurately approximated. Hence, gradient $\mathbf{g}$ and Hessian $\mathbf{H}$ are always computable. Meanwhile, because Eq. (3) is a convex optimization, the Newton method (Boyd & Vandenberghe, 2004) could be considered to address the optimization. However, classical Newton (or Quasi-Newton) methods cannot be directly used, because they are designed for unconstrained optimization (Shanno, 1970), which is not the case of Eq. (3).

Due to this, we adapt the projection Newton method (Bertsekas, 1982) to solve the optimization problem defined in Eq. (3). As outlined in Alg. 1, the iteration begins with the initial point $\mathbf{z}_{(0)}^\xi$. In each iteration $t$, we compute the gradient $\mathbf{g}_{(t)}$, Hessian $\mathbf{H}_{(t)}$, step size $\alpha_{(t)}$, and construct $\mathbf{D}_{(t)}$ as a partial diagonal approximation of $\mathbf{H}_{(t)}^{-1}$. After performing a Newton-like step, the point is projected into the feasible domain $[\mathbf{z}_{\text{left}}, \mathbf{z}_{\text{right}}]$. The step size $\alpha_{(t)}$ is determined using an Armijo-like rule (Boyd & Vandenberghe, 2004). Detailed definitions of $\mathrm{DiagApprox}(\cdot)$, $\mathrm{Armijo}(\cdot)$, and $\mathrm{Project}[\mathbf{z}_{\text{left}}, \mathbf{z}_{\text{right}}](\cdot)$ are provided in Appx. C. For learning the structural model ($\hat{\boldsymbol{\theta}}$) from observational data, we employ the least square estimation (LSE) for parameters $\hat{\boldsymbol{\beta}}_j$, and estimating $\hat{\boldsymbol{\Sigma}}$ based on residuals. Note that the time complexity of the decision-making step, specifically lines 5 to 11, can be bounded by $\mathcal{O}(|\mathbf{z}^\xi|^3 \cdot L)$ ($L$ is iteration rounds), as the dominant time-consuming operation in each iteration is computing the inverse of the matrix $\mathbf{H}(t) \in \mathbb{R}^{|\mathbf{z}^\xi| \times |\mathbf{z}^\xi|}$.

As a second-order optimization method, the presented projection Newton method in Alg. 1 enjoys a superlinear convergence speed by the following Thm. 3.8, similar to the classical Newton method (Boyd & Vandenberghe, 2004).

**Algorithm 1** Projection Newton for selecting optimal action

**Input:** observed $\mathbf{x}$; parameter $\hat{\boldsymbol{\theta}}$; $\mathcal{S}(\mathbf{M}, \mathbf{d})$ under CARE

1: Compute matrices $\mathbf{A}, \mathbf{B}, \mathbf{C}$ by $\{\hat{\beta}_j\}_{j=1}^{|\mathbf{V}|} \in \hat{\boldsymbol{\theta}}$

2: $\ell(\mathbf{z}^\xi) = \sum_{i=1}^{|\mathbf{Y}|} -\log\{\Delta_i(\mathbf{z}^\xi)\}$     $\triangleright \Delta_i(\mathbf{z}^\xi)$ in Cor. 3.7

3: Initialize $\mathbf{z}_{(0)}^\xi \in [\mathbf{z}_{\text{left}}, \mathbf{z}_{\text{right}}]$

4: **for** $t = 0$ **to** $L-1$ **do**

5:     Compute $\mathbf{g}_{(t)} = \frac{\partial}{\partial \mathbf{z}^\xi} \ell(\mathbf{z}^\xi)\big|_{\mathbf{z}^\xi = \mathbf{z}_{(t)}^\xi}$

6:     Compute $\mathbf{H}_{(t)} = \frac{\partial^2}{\partial \mathbf{z}^\xi \partial \mathbf{z}^{\xi\mathsf{T}}} \ell(\mathbf{z}^\xi)\big|_{\mathbf{z}^\xi = \mathbf{z}_{(t)}^\xi}$

7:                 $\triangleright$ Computation formula in Appx. C

8:     $\mathbf{D}_{(t)} = \text{DiagAppx}(\mathbf{H}_{(t)}^{-1})$, $\alpha_{(t)} = \text{Armijo}\left(\mathbf{D}_{(t)}\right)$

9:     $\mathbf{z}_{(t+1)}^\xi = \underset{[\mathbf{z}_{\text{left}}, \mathbf{z}_{\text{right}}]}{\text{Project}}\left(\mathbf{z}_{(t)}^\xi - \alpha_{(t)}\mathbf{D}_{(t)}\mathbf{g}_{(t)}\right)$

10:     **if** $\|\mathbf{z}_{(t+1)}^\xi - \mathbf{z}_{(t)}^\xi\| < \delta$ **then**

11:        Set $\mathbf{z}_{(L)}^\xi = \mathbf{z}_{(t+1)}^\xi$ and **break**

12:     **end if**

13: **end for**

**Output:** the optimal decision alteration $\mathbf{z}_{(L)}^\xi$

---

**Theorem 3.8.** *Let $\mathbb{P}_{\text{A}}^\star$ denote the maximal AUF probability $\mathbb{P}(\mathbf{Y} \in \mathcal{S} \mid \hat{\boldsymbol{\theta}}, \mathbf{x}, Rh(\mathbf{z}^\xi))$ that can be achieved by alteration $\mathbf{z}^\xi \in [\mathbf{z}_{\text{left}}, \mathbf{z}_{\text{right}}]$, and let $\{\mathbf{z}_{(k)}^\xi\}_{k=1}^L$ denote the sequence outputted by Alg. 1. The probability sequence $\{\mathbb{P}(\mathbf{Y} \in \mathcal{S} \mid \hat{\boldsymbol{\theta}}, \mathbf{x}, Rh(\mathbf{z}_{(k)}^\xi))\}_{k=1}^L$ converges to $\mathbb{P}_{\text{A}}^\star$ with a superlinear convergence rate (at least quadratic).*

As guaranteed in Thm. 3.8, the optimal alteration that maximizes $\mathbb{P}_{\text{AUF}}$ can be efficiently selected by Alg. 1. Note that $\mathbb{P}_{\text{AUF}}$ in Thm. 3.8 conditioned on $\hat{\boldsymbol{\theta}}$ rather than $\boldsymbol{\theta}$, this is because true parameters of the SRM are practically not available. We can only estimate the parameters from data, and then try to select the optimal decision alteration based on $\hat{\boldsymbol{\theta}}$ rather than true $\boldsymbol{\theta}$. Hence, the guarantee in Thm. 3.8 designed for Alg. 1 is conditioned on $\hat{\boldsymbol{\theta}}$ as well, and if the $\hat{\boldsymbol{\theta}}$ used in Alg. 1 can be changed as true $\boldsymbol{\theta}$, $\mathbb{P}_{\text{AUF}}$ in Thm. 3.8 also turns to condition on $\boldsymbol{\theta}$.

### 3.3. AUF scenarios beyond the CARE condition

Another crucial concern is that AUF scenarios exist beyond the CARE condition, where the transformation in Cor. 3.7 no longer holds. Analyzing the AUF probability in such cases is challenging due to irregular desired regions. To address this, we propose *inner CARE embedding*, which identifies an inner canonical rectangle $\mathcal{I}_{cr}$ for the desired region $\mathcal{S}$, such that $\mathbb{P}(\mathbf{Y} \in \mathcal{I}_{cr} \mid \hat{\boldsymbol{\theta}}, \mathbf{x}, Rh(\mathbf{z}^\xi))$ forms a strict lower bound for $\mathbb{P}(\mathbf{Y} \in \mathcal{S} \mid \hat{\boldsymbol{\theta}}, \mathbf{x}, Rh(\mathbf{z}^\xi))$. This ensures the transformation in Cor. 3.7 holds for the lower bound, allowing us to optimize it as a surrogate using Alg. 1. The formal definition of inner CARE embedding is in Def. 3.9.

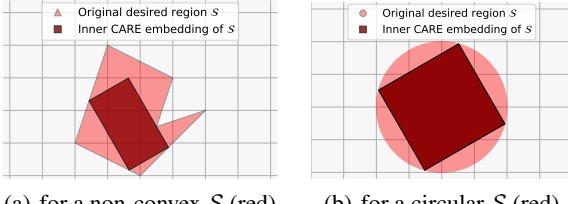

(a) for a non-convex $\mathcal{S}$ (red)      (b) for a circular $\mathcal{S}$ (red)

*Figure 3.* Examples of inner CARE embedding for different regions $\mathcal{S}$ with the same covariance matrix $\mathbf{C}^\mathsf{T}\boldsymbol{\Sigma}\mathbf{C}$

**Definition 3.9** (**Inner CARE embedding**)**.** Let $\mathcal{S}$ denote a non-empty region in $\mathbb{R}^d$, and let $\mathcal{I}_{cr}(\mathbf{M}, \mathbf{d})$ denote a convex polytope as defined in Def. 3.1. Then $\mathcal{I}_{cr}(\mathbf{M}, \mathbf{d})$ is called the inner CARE embedding of $\mathcal{S}$ iff the following holds:

(1) $\mathcal{I}_{cr}(\mathbf{M}, \mathbf{d})$ is a canonical rectangle *w.r.t.* $\mathbf{C}\boldsymbol{\Sigma}\mathbf{C}^\mathsf{T} \in \mathbb{R}^{d \times d}$, where $\mathbf{C}$ and $\boldsymbol{\Sigma}$ are defined in Lemma 3.4; and

(2) $\mathcal{I}_{cr}$ is the lagest subset of $\mathcal{S}$ satisfying (1) above.

With Def. 3.9, we can always define a region $\mathcal{I}_{cr}(\mathbf{M}, \mathbf{d})$ that satisfies the CARE condition for any non-empty desired region $\mathcal{S}$. As illustrated in Fig. 3, we shade the inner CARE embedding in dark red for some irregular desired regions $\mathcal{S}$, including a non-convex polytope as in Fig. 3(a) and a circular region as in Fig. 3(b). As the inner CARE embedding $\mathcal{I}_{cr}$ always satisfies the CARE condition, leveraging the transformation in Cor. 3.7 could be considered, and the following Prop. 3.10 guarantees its feasibility.

**Proposition 3.10.** *Let $\mathcal{S}$ denote a non-empty region in $\mathbb{R}^d$, and let $\mathcal{I}_{cr}(\mathbf{M}, \mathbf{d})$ denote the inner CARE embedding of $\mathcal{S}$. Then, it always holds that $\ell(\mathbf{z}^\xi) \leq \hat{\ell}(\mathbf{z}^\xi)$, where the loss function $\ell(\mathbf{z}^\xi) = -\log\left(\mathbb{P}(\mathbf{Y} \in \mathcal{S} \mid \hat{\boldsymbol{\theta}}, \mathbf{x}, Rh(\mathbf{z}^\xi))\right)$ and $\hat{\ell}(\mathbf{z}^\xi) = -\log\left(\mathbb{P}(\mathbf{Y} \in \mathcal{I}_{cr} \mid \hat{\boldsymbol{\theta}}, \mathbf{x}, Rh(\mathbf{z}^\xi))\right)$.*

Prop. 3.10 guarantees that for any irregular desired region $\mathcal{S}$ in the AUF problem, the inner CARE embedding method provides a strict upper bound $\hat{\ell}(\mathbf{z}^\xi)$ on the original loss function $\ell(\mathbf{z}^\xi)$. This upper bound serves as an effective surrogate optimization target, analogous to classical empirical risk minimization (ERM) (Mohri et al., 2018). Since inner CARE embedding always produces a region satisfying the CARE condition and enabling the convex transformation in Cor. 3.7, we can transform the intractable problem of optimizing $\mathbb{P}_{\text{AUF}}$ without an analyzable formula into the more manageable task of finding the inner CARE embedding for $\mathcal{S}$. It is worth noting that while the inner CARE embedding requires the largest subset of $\mathcal{S}$ satisfying the CARE condition, Prop. 3.10 remains valid for any CARE-compliant subset of $\mathcal{S}$. Finally, given that desired regions can vary substantially across applications, different inner CARE embedding methods need to be specifically designed for each distinct region type. Below, we provide a method for finding the inner CARE embedding for a circular region.

**Proposition 3.11.** *Let $\mathcal{S} = \{\mathbf{x} \in \mathbb{R}^d \mid \|\mathbf{x} - \mathbf{o}\|_2^2 = r^2\}$ denote a circular region on $\mathbb{R}^d$, and let $\mathbf{C}, \mathbf{\Sigma}$ denote problem-related matrices as defined in Lemma 3.4. Then, the inner CARE embedding of $\mathcal{S}$ can be expressed by $\mathcal{I}_{cr}(\mathbf{M}, \mathbf{d})$ with:*

$$
\begin{cases}
\mathbf{M} = (\mathbf{I}, -\mathbf{I})^\mathsf{T} \mathbf{\Lambda} \mathbf{Q} \\
\mathbf{d} = (\mathbf{I}, \mathbf{I})^\mathsf{T} \mathbf{\Lambda} \mathbf{1} \cdot r/\sqrt{d} + \mathbf{Mo},
\end{cases} \tag{4}
$$

*where $\mathbf{Q}^\mathsf{T} \mathbf{\Lambda} \mathbf{Q}$ is the eigen-decomposition of matrix $\mathbf{C} \mathbf{\Sigma} \mathbf{C}^\mathsf{T}$, $\mathbf{I} \in \mathbb{R}^{d \times d}$ is the identity matrix and $\mathbf{1} \in \mathbb{R}^{d \times 1}$ is the vector with all components equals 1.*

From Prop. 3.11 presented above, for a circular desired region $\mathcal{S}$, we can directly apply Eq. (4) to obtain its inner CARE embedding $\mathcal{I}_{cr}(\mathbf{M}, \mathbf{d})$. Although the original loss function, *i.e.*, $\ell(\mathbf{z}^\xi) = -\log\left(\mathbb{P}(\mathbf{Y} \in \mathcal{S} \mid \hat{\boldsymbol{\theta}}, \mathbf{x}, Rh(\mathbf{z}^\xi))\right)$, is generally not analyzable in this situation, Prop. 3.10 guarantees that we can optimize its analyzable upper bound, *i.e.*, $-\log\left(\mathbb{P}(\mathbf{Y} \in \mathcal{I}_{cr} \mid \hat{\boldsymbol{\theta}}, \mathbf{x}, Rh(\mathbf{z}^\xi))\right)$, as a surrogate by using the convex transformation defined in Cor. 3.7.

# 4. A closed-form solution for $|\mathbf{Y}| = 1$

In this section, we propose a closed-form solution for the formulated AUF problem in cases where the dimension of the concerned output equals 1 ($|\mathbf{Y}| = 1$) and the desired region of $\mathbf{Y}$ is an interval on $\mathbb{R}$. The proposed closed-form solution can greatly reduce the time complexity for selecting optimal alterations that maximize the AUF probability.

Considering situations where the output dimension is singleton, *i.e.*, $|\mathbf{Y}| = 1$, it can be verified that any interval region $\mathcal{S}$ on $\mathbb{R}$ satisfies the CARE condition. In this case, recall from Thm. 3.6 that the AUF probability can always be expressed as $\mathbb{P}_{\text{AUF}} = \Phi\left(-\mathbf{k}_1^\mathsf{T} \mathbf{z}^\xi - b_1\right) - \Phi\left(\mathbf{k}_2^\mathsf{T} \mathbf{z}^\xi + b_2\right)$, where $\mathbf{k}_i^\mathsf{T}$ is the $i$-th row of matrix $\mathbf{MB} \in \mathbb{R}^{2 \times |\mathbf{z}^\xi|}$ and $b_i$ is the $i$-th component of vector $\mathbf{MAx} - \mathbf{d} \in \mathbb{R}^{2 \times 1}$. Here, $\mathbf{M}$, $\mathbf{A}, \mathbf{B}, \mathbf{d}$, and $\mathbf{x}$ are defined in Lemma 3.4 and Thm. 3.6. Furthermore, according to the discussion in Remark 3.3, we can assume without loss of generality that $\mathbf{k}_i = -\mathbf{k}_{i+|\mathbf{Y}|}$, leading to $\mathbb{P}_{\text{AUF}} = \Phi\left(\mathbf{k}_2^\mathsf{T} \mathbf{z}^\xi - b_1\right) - \Phi\left(\mathbf{k}_2^\mathsf{T} \mathbf{z}^\xi + b_2\right)$. Remark that $-b_1 > b_2$ always holds because $\Phi(\cdot)$ is a monotone increasing function and $\mathbb{P}_{\text{AUF}} > 0$ as guaranteed in Thm. 3.6.

Using Alg. 1 to select the optimal alteration is also available in this case, which has a $\mathcal{O}(|\mathbf{z}^\xi|^3 \cdot L)$ time complexity as discussed in Sec. 3.2. To decrease the executing time by leveraging the information of $|\mathbf{Y}| = 1$, directly analyzing the gradient is worth considering. Specifically, the gradient of $\ell(\mathbf{z}^\xi) = -\log\left\{\Phi\left(\mathbf{k}_2^\mathsf{T} \mathbf{z}^\xi - b_1\right) - \Phi\left(\mathbf{k}_2^\mathsf{T} \mathbf{z}^\xi + b_2\right)\right\}$ in Eq. (3) can be derived as follows:

$$
\frac{\partial}{\partial \mathbf{z}^\xi} \ell(\mathbf{z}^\xi) = \frac{\phi\left(\mathbf{k}_2^\mathsf{T} \mathbf{z}^\xi + b_2\right) - \phi\left(\mathbf{k}_2^\mathsf{T} \mathbf{z}^\xi - b_1\right)}{\Phi\left(\mathbf{k}_2^\mathsf{T} \mathbf{z}^\xi - b_1\right) - \Phi\left(\mathbf{k}_2^\mathsf{T} \mathbf{z}^\xi + b_2\right)} \mathbf{k}_2. \tag{5}
$$

---

**Algorithm 2** Closed-form solution for cases where $|\mathbf{Y}| = 1$

**Input:** observed $\mathbf{x}$; parameter $\hat{\boldsymbol{\theta}}$; $\mathcal{S}(\mathbf{M}, \mathbf{d})$ under CARE

1: Compute matrices $\mathbf{A}, \mathbf{B}, \mathbf{C}$ by $\{\hat{\boldsymbol{\beta}}_j\}_{j=1}^{|\mathbf{V}|} \in \hat{\boldsymbol{\theta}}$
2: $\mathbf{K} = \mathbf{MB} \in \mathbb{R}^{2 \times |\mathbf{z}^\xi|}$; $\mathbf{b} = \mathbf{MAx} - \mathbf{d} \in \mathbb{R}^{2 \times 1}$
3: $\mathbf{k}^\mathsf{T} \leftarrow$ the 2nd row vector of matrix $\mathbf{K}$
4: $b \leftarrow (b_1 - b_2)/2$     ▷ $b_i$ is the $i$-th component of $\mathbf{b}$
5: Select $\mathbf{z}^\xi = \text{CLOSED-SOLUTION}(\mathbf{k}, b, \mathbf{z}_{\text{left}}, \mathbf{z}_{\text{right}})$
6: **function** CLOSED-SOLUTION($\mathbf{k}, b, \boldsymbol{l}, \boldsymbol{r}$)
7:     $m = \mathbf{k}^\mathsf{T} \cdot (\mathbb{I}(\mathbf{k} \geq \mathbf{0}) \circ \boldsymbol{l} + \mathbb{I}(\mathbf{k} < \mathbf{0}) \circ \boldsymbol{r})$
8:     $M = \mathbf{k}^\mathsf{T} \cdot (\mathbb{I}(\mathbf{k} < \mathbf{0}) \circ \boldsymbol{l} + \mathbb{I}(\mathbf{k} \geq \mathbf{0}) \circ \boldsymbol{r})$
9:     **if** $b \leq m$ **then**
10:        $\mathbf{z}^\star = \mathbb{I}(\mathbf{k} \geq \mathbf{0}) \circ \boldsymbol{l} + \mathbb{I}(\mathbf{k} < \mathbf{0}) \circ \boldsymbol{r}$
11:     **else if** $b \geq M$ **then**
12:        $\mathbf{z}^\star = \mathbb{I}(\mathbf{k} < \mathbf{0}) \circ \boldsymbol{l} + \mathbb{I}(\mathbf{k} \geq \mathbf{0}) \circ \boldsymbol{r}$
13:     **else** $\mathbf{z}^\star \leftarrow \mathbf{0}$, and **do**
14:        **for** $j = 0$ **to** $|\mathbf{z}^\star|$ **do**
15:           **if** $k_j = 0$ **then**
16:              **continue**
17:           **else if** $b/k_j \in [l_j, r_j]$ **then**
18:              $z_j^\star = b/k_j$ and **break**
19:           **else**
20:              $z_j^\star = \mathbb{I}(b/k_j < l_j)l_j + \mathbb{I}(b/k_j \geq r_j)r_j$
21:              $b \leftarrow b - z_j^\star$
22:           **end if**
23:        **end for**
24:     **end if**
25: **return** $\mathbf{z}^\star$

**Output:** the optimal decision alteration $\mathbf{z}^\xi$

---

Intuitively, directly analyzing the stationary points of $\ell(\mathbf{z}^\xi)$ can reduce execution time. As shown in Thm. 3.6, $\ell(\mathbf{z}^\xi)$ is convex *w.r.t.* $\mathbf{z}^\xi$, so any stationary point $\mathbf{z}^\star$ of $\ell(\mathbf{z}^\xi)$, *i.e.*, where $\frac{\partial}{\partial \mathbf{z}^\xi} \ell(\mathbf{z}^\xi)\big|_{\mathbf{z}^\xi = \mathbf{z}^\star} = \mathbf{0}$, achieves the global minimum. In cases where $|\mathbf{Y}| = 1$, this idea is feasible, as Alg. 2 constructively provides a zero point $\mathbf{z}^\star$ of Eq. (5).

As shown in Alg. 2, the function CLOSED-SOLUTION primarily utilizes *if-else* statements and iterates over the vector $\mathbf{z}^\star$. Therefore, this function can be represented in a closed form as a complex equation, which depends on $\mathbf{k}, b$, $\boldsymbol{l}$, and $\boldsymbol{r}$ using the indicator function $\mathbb{I}(\cdot)$ and the summation operator $\sum$. Specifically, Alg. 2 focuses on finding the root for Eq. (5), which suffices to be an optimal alteration that achieves the maximal AUF probability. Recognizing that $\mathbf{k}_2$ does not necessarily equal $\mathbf{0}$, the point $\mathbf{z}$ is a root of Eq. (5) if and only if $\phi\left(\mathbf{k}_2^\mathsf{T} \mathbf{z} - b_1\right) = \phi\left(\mathbf{k}_2^\mathsf{T} \mathbf{z} + b_2\right)$. Given the even nature of $\phi(\cdot)$, which is monotonic on $(-\infty, 0]$, and the fact that $-b_1 > b_2$ as previously discussed, it follows that $\mathbf{k}_2^\mathsf{T} \mathbf{z} - b_1 = -\mathbf{k}_2^\mathsf{T} \mathbf{z} - b_2$, which simplifies to $\mathbf{k}_2^\mathsf{T} \mathbf{z} = \frac{b_1 - b_2}{2}$. Alg. 2 is designed to solve this equation under the constraint $\mathbf{z} \in [\mathbf{z}_{\text{left}}, \mathbf{z}_{\text{right}}]$, where $M$ and $m$ represent the maximum and minimum values achievable by $\mathbf{k}_2^\mathsf{T} \mathbf{z}$ under this constraint. If there exists a $\mathbf{z}^\star \in [\mathbf{z}_{\text{left}}, \mathbf{z}_{\text{right}}]$ such that

$\mathbf{k}_2^\mathsf{T}\mathbf{z}^\star = \frac{b_1 - b_2}{2}$, this root is selected as shown in lines 14 to 20; otherwise, the boundary value is chosen as described in lines 10 to 13. Further discussions are provided in Appx. B, along with the proof for the following Thm. 4.1.

**Theorem 4.1.** *When* $|\mathbf{Y}| = 1$*, let* $\mathbb{P}_A^\star$ *denote the maximal AUF probability* $\mathbb{P}(Y \in [y_l, y_r] \mid \hat{\boldsymbol{\theta}}, \mathbf{x}, Rh(\mathbf{z}^\xi))$ *that can be achieved by alteration* $\mathbf{z}^\xi \in [\mathbf{z}_{\text{left}}, \mathbf{z}_{\text{right}}]$*, and let* $\mathbf{z}^\star$ *denote the alteration outputted by Alg. 2. Then the probability* $\mathbb{P}(Y \in [y_l, y_r] \mid \hat{\boldsymbol{\theta}}, \mathbf{x}, Rh(\mathbf{z}^\star))$ *equals* $\mathbb{P}_A^\star$*.*

As guaranteed in Thm. 4.1, the optimal alteration that maximizes $\mathbb{P}_{\text{AUF}}$ can be effectively determined by Alg. 2. The issue of conditioning on $\hat{\boldsymbol{\theta}}$ rather than true $\boldsymbol{\theta}$ follows the same discussion as Thm. 3.8. It is worth noting that the time complexity of the decision-making step in this case, *i.e.*, the time complexity of the function in Alg. 2, is bounded by $\mathcal{O}(|\mathbf{z}^\xi|)$, because all operations can be reduced to iterating over elements of vectors with length $|\mathbf{z}^\xi|$, including $\mathbf{k}$, $\boldsymbol{l}$, $\boldsymbol{r}$, and $\mathbf{z}^\star$. Compared with Alg. 1, which has a time complexity of $\mathcal{O}(|\mathbf{z}^\xi|^3 \cdot L)$, the execution time for selecting the optimal alteration in Alg. 2 is greatly accelerated by leveraging the information that $|\mathbf{Y}| = 1$. Last but not least, we would like to emphasize that $|\mathbf{Y}| = 1$ is a crucial case, as most of the experiments in existing rehearsal learning research are established on this scenario (Qin et al., 2023; Du et al., 2024). Therefore, developing a theoretically efficient approach for this case is essential for practical applications.

## 5. Discussion

Our approach aims to select optimal alteration $\mathbf{z}^\xi$ that maximizes the AUF probability $\mathbb{P}_{\text{AUF}}$. Under the CARE condition, it precisely determines the optimal alteration. The condition offers practical advantages, as decision-makers can strategically design the desired region $\mathcal{S}$ to satisfy the CARE condition after estimating system parameters. For scenarios beyond CARE, our approach constructs inner CARE embeddings. While this construction remains challenging for arbitrary regions, we demonstrate its feasibility for specific geometries such as circular regions in Prop. 3.11. For other region types, established techniques like maximum rectangle algorithms for polygons (Choi et al., 2021) provide powerful tools, and future advances in these methods will further expand our approach's applicability.

Meanwhile, while our theoretical analysis is established with the Gaussian noise assumption that is widely adopted across various domains (Shumway et al., 2000; Cohen et al., 2018), our methodology remains effective in general cases through appropriate Gaussian approximations. For instance, Laplace's approximation (Bishop & Nasrabadi, 2006) in Bayesian statistics demonstrates how unimodal distributions can be effectively approximated by fitting a Gaussian distribution centered at their mode.

## 6. Related work

We discuss two types of works related to our study, including reinforcement learning (RL) and causality:

**Reinforcement Learning (RL).** RL methods have demonstrated remarkable success across various decision-making domains (Sutton & Barto, 2018). In general, classical RL approaches (Lillicrap et al., 2016; Haarnoja et al., 2018) typically require extensive cost-free interactions, which are unavailable in AUF setting. While numbers of offline RL (Li et al., 2023; Qiao & Wang, 2023) and hybrid offline-online RL (Song et al., 2022; Pong et al., 2022) methods have been developed to mitigate interaction requirements, they remain ill-suited for AUF scenarios (which demands immediate decisions *without* interactions) and necessitate substantial offline datasets. Furthermore, when using MDP to model AUF, the reward function may differ substantially between offline and online data, as the distribution of $\mathbf{Y}$ given observed $\mathbf{z}$ ($\mathbb{P}(\mathbf{Y} \mid \mathbf{z})$) can fundamentally differ from that under alterations on $\mathbf{z}$ ($\mathbb{P}(\mathbf{Y} \mid Rh(\mathbf{z}))$), resulting in mismatched reward functions $\mathbb{I}(\mathbf{Y} \in \mathcal{S} \mid \mathbf{z})$ versus $\mathbb{I}(\mathbf{Y} \in \mathcal{S} \mid Rh(\mathbf{z}))$. In contrast, rehearsal learning leverages structural relations among variables, enabling it to recommend immediate decisions based solely on observational data without requiring interactions. Note that for addressing AUF, finding influence relations is a key step, while RL could be a helpful tool for finding the relations with proper adaptations.

**Causality.** Extensive research has explored structural models for decision-making, primarily based on structural causal models (Spirtes et al., 2000). This work encompasses two notable categories: (i) methods for identifying causal structures or effects (Kocaoglu et al., 2017; Wang et al., 2020; 2023b; Qin et al., 2021; Wang et al., 2023a; Zhang et al., 2023; Qin et al., 2024), which focus primarily on causal discovery and estimation; and (ii) approaches for determining optimal intervention points using causal bandits or causal RL methods (Lattimore et al., 2016; Sen et al., 2017; Lee & Bareinboim, 2018; Zhang & Bareinboim, 2019; Lu et al., 2021; Park et al., 2025), which incorporate additional utility considerations. These methods generally rely on causal modeling, which may be luxurious or restrictive in some real-world decision-making scenarios (Zhou, 2022). The strong assumptions required for causal identification often limit the feasibility of finding viable alterations. Conversely, correlation-based approaches, which underpin most ML models, are generally not sufficient for decision-making. Based on this insight, rehearsal is proposed (Zhou, 2022), which captures influence relations for addressing the AUF problem (Zhou, 2023). Qin et al. (2023) further propose the SRM and rehearsal-learning framework, which can adapt to dynamically evolving decision systems. Note that this framework can recommend effective alterations across both time-varying systems and scenarios with mutual influences.

# 7. Experiments

We evaluate our proposed approach on two datasets including a synthetic dataset and a real-world dataset. We compare our method with two baseline methods and the existing rehearsal learning methods, including QWZ23 (Qin et al., 2023) and AUF-MICNS (Du et al., 2024). Besides, although we have discussed in Sec. 6 that RL methods are not well-suited for the AUF problem due to limited or even nonexistent interactions with the decision environment, we still implement some classic RL algorithms for comparison to mitigate potential concerns. The compared RL algorithms includes DDPG (Lillicrap et al., 2016), PPO (Schulman et al., 2017), and SAC (Haarnoja et al., 2018). First, we briefly introduce the datasets, with details listed in Appx. D.

**Synthetic Data (Manage).** The Manage dataset is designed to simulate a market management scenario where a market manager must make decisions to promote total profit (TPF) and customer numbers (NCT). Key variables affecting TPF and NCT include the cost of raw materials (C), product price (P), and competitor's price (E), among others. We assume that P and C are the actionable variables in this context. There are mutually influenced relationships among these variables: setting P low leads to a competitive drop in E, and vice versa. The dimensions of $\mathbf{X}$, $\mathbf{Z}$, and $\mathbf{Y}$ are 2, 4, and 2, respectively. The desired region $\mathcal{S}$ for $\mathbf{Y}$ is a circular region, and over 95% of value $\mathbf{Y}$ fails to fall within $\mathcal{S}$ in natural conditions as shown in Fig. 5(a).

**Real-world Data (Bermuda).** The Bermuda dataset, which records environmental variables in the Bermuda area, is described in ecology research (Courtney et al., 2017), with available generation order of variables (Andersson & Bates, 2018). The dimensions of $\mathbf{X}$, $\mathbf{Z}$, and $\mathbf{Y}$ are 3, 7, and 1, and the desired region $\mathcal{S}$ for $\mathbf{Y}$ is $\mathcal{S} = \{\text{NEC} \in [0.5, 2]\}$. In natural conditions, over 90% of value $\mathbf{Y}$ fails to fall within $\mathcal{S}$, as shown in Fig. 5(a). The parameters of structural equations are derived from fitting linear models on normalized data (Qin et al., 2023). It is assumed that 5 variables in $\mathbf{Z}$ are actionable (Aglietti et al., 2020), and feasible alteration values are set to $[-1, 1]$ for each of them.

Meanwhile, we repeat the experiment under 100 random seeds for each dataset, incluing 3 measures as follows:

1. **AUF Prob. given x.** The probability that the selected action can successfully make $\mathbf{Y} \in \mathcal{S}$ (conditioned on x and $\boldsymbol{\theta}$). This is an immediate decision, and the probability is approximated by Monte Carlo method.
2. **100-Rds AUF Freq.** Success counts of making $\mathbf{Y} \in \mathcal{S}$ in a 100-round experiment (with different observations xs). This is a sequential decision-making scenario, in which the RL methods could update the policy.
3. **Avg. Time.** Average executing time for making one decision (the time only for the decision-making step).

*Table 2.* Results of synthetic data (mean ± std).

| METHOD | AUF PROB.(%) | 100-RDS SUCC. FREQ. | AVG. TIME (MS) |
|---|---|---|---|
| NO ACTION | $2.52 \pm 9.19$ | $1.61 \pm 1.29$ | \ |
| RANDOM | $0.97 \pm 7.01$ | $1.60 \pm 1.52$ | \ |
| DDPG | $1.44 \pm 9.53$ | $1.57 \pm 1.16$ | $6.35 \pm 0.28$ |
| SAC | $0.98 \pm 8.07$ | $1.55 \pm 1.16$ | $10.85 \pm 0.22$ |
| PPO | $1.42 \pm 8.02$ | $1.65 \pm 1.13$ | $11.50 \pm 1.84$ |
| QWZ23 | $94.22 \pm 1.25$ | $94.06 \pm 1.25$ | $85.37 \pm 69.3$ |
| MICNS | $96.96 \pm 8.12$ | $96.99 \pm 8.44$ | $\mathbf{4.02} \pm 1.28$ |
| **OURS** | $\mathbf{99.01} \pm 0.62$ | $\mathbf{98.93} \pm 1.26$ | $5.86 \pm 2.17$ |

*Table 3.* Results of Bermuda data (mean ± std).

| METHOD | AUF PROB.(%) | 100-RDS SUCC. FREQ. | AVG. TIME (MS) |
|---|---|---|---|
| NO ACTION | $8.77 \pm 3.36$ | $8.94 \pm 2.76$ | \ |
| RANDOM | $15.47 \pm 20.9$ | $14.93 \pm 3.41$ | \ |
| DDPG | $14.46 \pm 27.6$ | $20.70 \pm 5.52$ | $15.17 \pm 3.57$ |
| SAC | $14.05 \pm 31.1$ | $18.21 \pm 3.94$ | $12.48 \pm 0.66$ |
| PPO | $15.91 \pm 33.8$ | $17.15 \pm 3.71$ | $13.31 \pm 2.59$ |
| QWZ23 | $73.47 \pm 7.11$ | $72.62 \pm 4.21$ | $567.7 \pm 129$ |
| MICNS | $76.19 \pm 20.8$ | $76.69 \pm 14.3$ | $23.61 \pm 13.2$ |
| **OURS** | $\mathbf{82.76} \pm 4.44$ | $\mathbf{83.26} \pm 3.68$ | $\mathbf{0.91} \pm 0.07$ |

The comparison results are summarized in Tab. 2 and Tab. 3, where the number of observational samples is set to 100. For immediate decisions, RL methods rely on default policies, resulting in performance similar to random actions. With limited interactions, RL methods can gather information to update their policies but still fail to produce actions comparable to rehearsal methods due to limited interactions. In contrast, rehearsal methods can generate effective actions by exploiting structural relations among environment variables. Our approach consistently outperforms others across all datasets under the AUF probability measure, which is the core concern of AUF problem. In terms of execution time, our approach has the same complexity ($\mathcal{O}(|\mathbf{z}|^3)$) as AUF-MICNS when $|\mathbf{Y}| > 1$, but achieves efficient decision-making with a reduced complexity of $\mathcal{O}(|\mathbf{z}|)$ for $|\mathbf{Y}| = 1$, leveraging the closed-form solution provided in Alg. 2. Finally, the hyperparameter $\tau$ for previous rehearsal-learning methods is selected as the value that achieves the highest average AUF probability among various candidates.

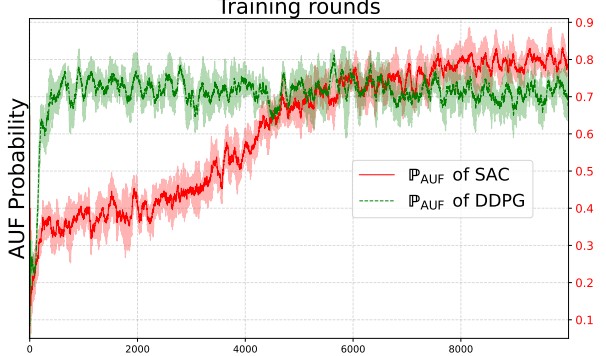

*Figure 4.* RL convergence curves (DDPG and SAC) on Bermuda data with adequate number of interactions.

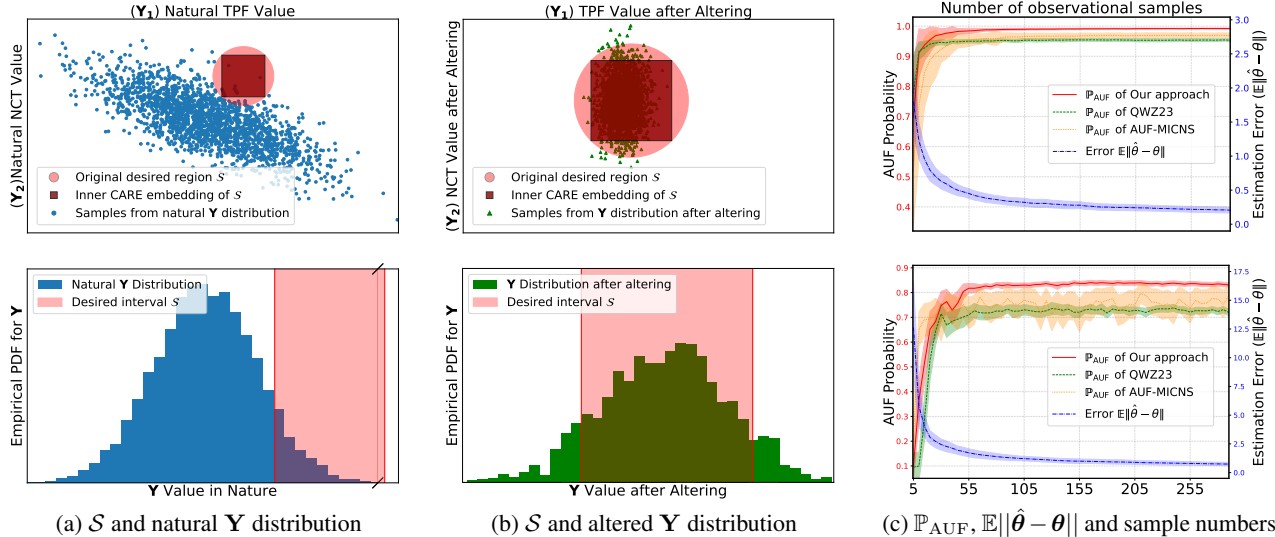

(a) $\mathcal{S}$ and natural $\mathbf{Y}$ distribution  (b) $\mathcal{S}$ and altered $\mathbf{Y}$ distribution  (c) $\mathbb{P}_{\mathrm{AUF}}$, $\mathbb{E}||\hat{\boldsymbol{\theta}} - \boldsymbol{\theta}||$ and sample numbers

*Figure 5.* Results for the synthetic dataset (row 1) and Bermuda dataset (row 2), respectively. For each dataset, the desired region shaded in red in column (a) is identical to that in column (b). The bands in column (c) represent the standard deviations.

Fig. 4 illustrates the convergence curves of DDPG and SAC on Bermuda data, demonstrating that classical RL methods perform well with sufficient interactions, despite their poor performance in the AUF setting where interactions are limited (Tab. 3). Fig. 5 presents: (a) the original distribution of outcome $\mathbf{Y}$; (b) the distribution of $\mathbf{Y}$ after implementing alterations; and (c) the relationship between AUF probability $\mathbb{P}_{\mathrm{AUF}}$, estimation error $\mathbb{E}||\hat{\boldsymbol{\theta}} - \boldsymbol{\theta}||$, and the number of *observational samples*. As evident in Fig. 5(b), our proposed approach successfully shifts the distribution of $\mathbf{Y}$ toward the desired region $\mathcal{S}$ through selected alterations. Furthermore, Fig. 5(c) shows that our method converges to the optimal solution as the number of observational samples increases, without requiring any interactions with the environment.

## 8. Conclusion

Rehearsal learning provides a promising framework for addressing the AUF problem in practical applications. However, existing rehearsal methods cannot identify the optimal decision alterations that maximize the AUF probability. To overcome this limitation, we proposed the CARE condition, which enables the transformation of the AUF problem into a convex optimization. For solving this transformed problem, we developed the projection-Newton algorithm that guarantees optimal decisions with superlinear convergence. Recognizing that some AUF scenarios fall outside the CARE condition, we introduced inner CARE embedding to generalize our approach. Additionally, we proved that when $|\mathbf{Y}| = 1$, AUF problem admits a closed-form solution, enabling decision-making with time complexity $\mathcal{O}(|\mathbf{z}|)$, significantly improving efficiency over prior methods. Experiments validate the effectiveness of our approach.

## Acknowledgements

This research was supported by Jiangsu Science Foundation Leading-edge Technology Program (BK20232003), NSFC (62406137). Tian-Zuo Wang was supported by National Postdoctoral Program for Innovative Talent and Xiaomi Foundation. The authors thank Tian Qin for discussions.

## Impact Statement

This paper presents work whose goal is to advance the field of Machine Learning. There are many potential societal consequences of our work, none which we feel must be specifically highlighted here.

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

# A. Definitions

## A.1. Details about Structural Rehearsal Models

In this section, we provide comprehensive definitions and discussions of the Structural Rehearsal Model (SRM), a probabilistic graphical model introduced by (Qin et al., 2023) to characterize influence relations among variables in AUF problem. The original definitions and discussions of the SRM can be found in Qin et al. (2023), and in this paper, we do not consider the time $t$ since we focus on the immediate decision making.

**• Definition of the rehearsal graph**

**Definition A.1** (Mixed graph, (Qin et al., 2023))**.** Let $G = (\mathbf{V}, \mathbf{E})$ be a graph, where $\mathbf{V}$ denotes the vertices and $\mathbf{E}$ the edges. $G$ is a mixed graph if for any distinct vertices $u, v \in \mathbf{V}$, there is at most one edge connecting them, and the edge is either directional ( $u \to v$ or $u \leftarrow v$ ) or bi-directional ($u \leftrightarrow v$).

**Definition A.2** (Bi-directional clique, (Qin et al., 2023))**.** A bi-directional clique $C = (\mathbf{V}^c, \mathbf{E}^c)$ of a mixed graph $G = (\mathbf{V}, \mathbf{E})$ is a complete subgraph induced by $\mathbf{V}^c \subseteq \mathbf{V}$ such that any edge $e \in \mathbf{E}^c$ is bi-directional. $C$ is maximal if adding any other vertex does not induce a bi-directional clique.

**Definition A.3** (Rehearsal graph, (Qin et al., 2023))**.** Let $G = (\mathbf{V}, \mathbf{E})$ be a mixed graph. Let $\{C_i\}_{i=1}^l$ denote all maximal bi-directional cliques of $G$, where $C_i = (\mathbf{V}_i^c, \mathbf{E}_i^c)$. $G$ is a rehearsal graph if and only if:

1. $\mathbf{V}_i^c \cap \mathbf{V}_j^c = \emptyset$ for any $i \neq j$.

2. $\forall i \in [l]$, if there is any edge pointing from some $u \in \mathbf{V} \backslash \mathbf{V}_i^c$ to some $v \in \mathbf{V}_i^c$, then $\forall v \in \mathbf{V}_i^c, u \to v$.

3. There exists a topological ordering for $\{C_i\}_{i=1}^l$ following the directions of directional edges between $C_i$s.

Note that the topological ordering for bi-directional cliques $\{C_i\}_{i=1}^l$ in $G$ reflects the generation order of involved variables.

**• The associated structural equations**

Associated with the graphical representation, the structural equations are defined over the bi-directional cliques $\{C_i\}_{i=1}^l$. Specifically, these equations are parameterized by $\boldsymbol{\theta}$, which comprises the set of parameter matrices $\{\boldsymbol{\beta}_i\}$ and the covariance matrix $\boldsymbol{\Sigma}_i$ for each clique $C_i$ in the rehearsal graph $G$. Notably, the quantitative influence of a directed edge $A \to B$, where $A \in C_a$ and $B \in C_b$, is captured in the parameter matrix $\boldsymbol{\beta}_b$. This is due to the topological ordering of the bi-directional cliques reflecting the temporal sequence of the generation process. In contrast, the influence of a bi-directional edge $D_1 \leftrightarrow D_2$ for $D_1, D_2 \in C_d$ is encoded in the covariance matrix $\boldsymbol{\Sigma}_d$, as mutual influence between variables ceases once the system reaches equilibrium.

For clarity in the main paper, we simplify the definition of dynamic structural equations in Eq. (1) to operate at the variable level rather than the clique level. For each variable $V_j \in C_i$, the associated parameter vector can be extracted directly from the corresponding parameter matrices $\{\boldsymbol{\beta}_i\}$. Similarly, the variance $\sigma_j^2$ for each variable $V_j \in C_i$ can be obtained from the respective covariance matrix $\boldsymbol{\Sigma}_i$.

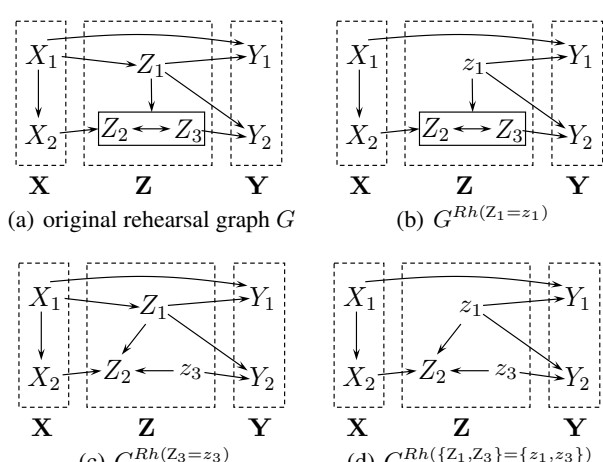

(a) original rehearsal graph $G$

(b) $G^{Rh(Z_1=z_1)}$

(c) $G^{Rh(Z_3=z_3)}$

(d) $G^{Rh(\{Z_1,Z_3\}=\{z_1,z_3\})}$

*Figure 6.* Fig. 6(a) is a rehearsal graph, Fig. 6(b)$\sim$ Fig. 6(d) is the corresponding alteration graphs with alterations. When an alteration occurs to certain variables, all incoming arrows to those variables are removed while other graph structures preserved.

Lastly, while the dynamic structural equations in Eq. (1) assume Gaussian noise for all $V_j \in \mathbf{V}$, it is worth emphasizing that for variables in cliques without parents, their noise terms can follow arbitrary distributions. This flexibility does not affect our proposed approach, as the columns of the matrix $\mathbf{C}$ corresponding to such variables are zero (as per Lemma 3.4). Consequently, both the theoretical results and the proposed algorithms remain unaffected.

# B. Proofs

In this section, we prove the asserts appeared in the paper.

## B.1. Proof of Theorem 3.6

**Lemma B.1.** *Let* $h(\mathbf{x}, \lambda) = -\log\left(\Phi\left(\mathbf{c}^\mathsf{T}\mathbf{x} + \lambda\right) - \Phi\left(\mathbf{c}^\mathsf{T}\mathbf{x}\right)\right)$ *where* $\lambda > 0$ *and* $\mathbf{c}$ *is a non-zero constant vector. Then* $h(\mathbf{x}, \lambda)$ *is convex w.r.t.* $\mathbf{x}$*, since the Hessian matrix* $\nabla_\mathbf{x}^2 h(\mathbf{x}, \lambda) \succeq 0$.

*Proof.* First, the gradient and Hessian *w.r.t.* $\mathbf{x}$ can be derived as:

$$\nabla_\mathbf{x} h = -\frac{\phi\left(\mathbf{c}^\mathsf{T}\mathbf{x} + \lambda\right) - \phi\left(\mathbf{c}^\mathsf{T}\mathbf{x}\right)}{\Phi\left(\mathbf{c}^\mathsf{T}\mathbf{x} + \lambda\right) - \Phi\left(\mathbf{c}^\mathsf{T}\mathbf{x}\right)} \cdot \mathbf{c} \tag{6}$$

$$\nabla_\mathbf{x}^2 h = \frac{\left(\Phi\left(\mathbf{c}^\mathsf{T}\mathbf{x} + \lambda\right) - \Phi\left(\mathbf{c}^\mathsf{T}\mathbf{x}\right)\right)\left\{\left(\mathbf{c}^\mathsf{T}\mathbf{x} + \lambda\right)\phi\left(\mathbf{c}^\mathsf{T}\mathbf{x} + \lambda\right) - \mathbf{c}^\mathsf{T}\mathbf{x}\phi\left(\mathbf{c}^\mathsf{T}\mathbf{x}\right)\right\} + \left(\phi\left(\mathbf{c}^\mathsf{T}\mathbf{x} + \lambda\right) - \phi\left(\mathbf{c}^\mathsf{T}\mathbf{x}\right)\right)^2}{\left(\Phi\left(\mathbf{c}^\mathsf{T}\mathbf{x} + \lambda\right) - \Phi\left(\mathbf{c}^\mathsf{T}\mathbf{x}\right)\right)^2} \cdot \mathbf{c}\mathbf{c}^\mathsf{T} \tag{7}$$

Let $f(x, \lambda) = (\Phi(x + \lambda) - \Phi(x))\{(x + \lambda)\phi(x + \lambda) - x\phi(x)\} + (\phi(x + \lambda) - \phi(x))^2$. Notice that in Eq. (7), $\left(\Phi\left(\mathbf{c}^\mathsf{T}\mathbf{x} + \lambda\right) - \Phi\left(\mathbf{c}^\mathsf{T}\mathbf{x}\right)\right)^2 > 0$ and $\mathbf{c}\mathbf{c}^\mathsf{T} \succeq 0$ always hold. Hence, it suffices to prove that $\nabla_\mathbf{x}^2 h \succeq 0$ if we can prove that $f(x, \lambda) > 0$ holds for $\forall x \in \mathbb{R}$ when $\lambda > 0$. Notice that:

$$f(x, \lambda) = \phi(x)f_1(x, \lambda) + \phi(x + \lambda)f_2(x, \lambda), \text{where}$$

$$\begin{cases} f_1(x, \lambda) = -x\left(\Phi(x + \lambda) - \Phi(x)\right) + \phi(x) - \phi(x + \lambda) \\ f_2(x, \lambda) = (x + \lambda)\left(\Phi(x + \lambda) - \Phi(x)\right) + \phi(x + \lambda) - \phi(x). \end{cases}$$

Since $\phi(x) > 0$ and $\phi(x + \lambda) > 0$ always holds, thus when $\lambda > 0$, if we can prove that $f_1(x, \lambda), f_2(x, \lambda) > 0$ holds for $\forall x \in \mathbb{R}$, it suffices to prove that $f(x, \lambda) > 0$ holds for $\forall x \in \mathbb{R}$. Notice that when $\lambda > 0$, it can be derived that:

$$\begin{cases} \nabla_\lambda f_1 = -x\phi(x + \lambda) + (x + \lambda)\phi(x + \lambda) = \lambda\phi(x + \lambda) > 0 \\ \nabla_\lambda f_2 = \Phi(x + \lambda) - \Phi(x) + (x + \lambda)\phi(x + \lambda) - (x + \lambda)\phi(x + \lambda) = \Phi(x + \lambda) - \Phi(x) > 0. \end{cases}$$

Hence, for $\forall x \in \mathbb{R}$, $f_1(x, \lambda) > f_1(x, 0) = 0$ and $f_2(x, \lambda) > f_2(x, 0) = 0$, which suffices to prove that $\nabla_\mathbf{x}^2 h \succeq 0$. □

**Theorem 3.6.** *We call the desired region* $\mathcal{S}(\mathbf{M}, \mathbf{d})$ *satisfies the CARE condition iff* $\mathcal{S}(\mathbf{M}, \mathbf{d})$ *is canonical w.r.t.* $\mathbf{C\Sigma C}^\mathsf{T}$*. Let* $\mathbb{P}_\mathrm{A}$ *denote* $\mathbb{P}(\mathbf{Y} \in \mathcal{S} \mid \boldsymbol{\theta}, \mathbf{x}, Rh(\mathbf{z}^\xi))$*,* $\mathbf{k}_j^\mathsf{T}$ *denote the* $j$*-th row of* $\mathbf{MB}$ *and* $b_j$ *denote the* $j$*-th component of* $\mathbf{MAx} - \mathbf{d}$*, the following asserts hold under the CARE condition:*

1. *Let* $|\mathbf{Y}| = d$*, the AUF probability can be expressed as*

$$\mathbb{P}_\mathrm{A} = \prod_{i=1}^{d}\left\{\Phi\left(-\mathbf{k}_i^\mathsf{T}\mathbf{z}^\xi - b_i\right) - \Phi\left(\mathbf{k}_{i+d}^\mathsf{T}\mathbf{z}^\xi + b_{i+d}\right)\right\};$$

2. *For any bounded alteration* $\mathbf{z}^\xi$*,* $\mathbb{P}_\mathrm{A} > 0$ *and the function* $\ell\left(\mathbf{z}^\xi\right) = -\log\left(\mathbb{P}_\mathrm{A}\right)$ *is convex w.r.t.* $\mathbf{z}^\xi$ .

*Proof.* Since $\mathcal{S}(\mathbf{M}, \mathbf{d})$ is canonical *w.r.t.* $\mathbf{C\Sigma C}^\mathsf{T}$, as discussed in Remark 3.3, we define without loss of generality that:

$$\mathbf{M} = \begin{bmatrix} \mathbf{I} & -\mathbf{I} \end{bmatrix}^\mathsf{T} \mathbf{D}^{-\frac{1}{2}}\mathbf{Q} \in \mathbb{R}^{2p \times p},$$

where $p = |\mathbf{Y}|$, $\mathbf{C\Sigma C}^\mathsf{T} = \mathbf{Q}^\mathsf{T}\mathbf{DQ}$ is the eigen decomposition of matrix $\mathbf{C\Sigma C}^\mathsf{T}$. Meanwhile, from Lemma 3.4, we know that $\mathbf{Y} \sim \mathcal{N}\left(\mathbf{Ax} + \mathbf{Bz}^\xi, \mathbf{C\Sigma C}^\mathsf{T}\right)$. Hence, it can be derived that:

$$\begin{aligned} \mathbf{MY} \leq \mathbf{d} &\Leftrightarrow \mathbf{MY} - \mathbf{M}\left(\mathbf{Ax} + \mathbf{Bz}^\xi\right) \leq \mathbf{d} - \mathbf{M}\left(\mathbf{Ax} + \mathbf{Bz}^\xi\right) \\ &\Leftrightarrow \mathbf{M}\left(\mathbf{Y} - \mathbf{Ax} - \mathbf{Bz}^\xi\right) \leq \mathbf{d} - \mathbf{M}\left(\mathbf{Ax} + \mathbf{Bz}^\xi\right) \\ &\Leftrightarrow (\mathbf{I}, -\mathbf{I})^\mathsf{T}\mathbf{D}^{-\frac{1}{2}}\mathbf{Q}\left(\mathbf{Y} - \mathbf{Ax} - \mathbf{Bz}^\xi\right) \leq \mathbf{d} - \mathbf{M}\left(\mathbf{Ax} + \mathbf{Bz}^\xi\right) \\ &\Leftrightarrow (\mathbf{I}, -\mathbf{I})^\mathsf{T}\boldsymbol{\nu} \leq \mathbf{d} - \mathbf{M}\left(\mathbf{Ax} + \mathbf{Bz}^\xi\right), \text{where } \boldsymbol{\nu} \sim \mathcal{N}(\mathbf{0}, \mathbf{I}_{p \times p}) \end{aligned}$$

That is to say, it holds that:

$$\mathbb{P}(\mathbf{Y} \in \mathcal{S} \mid \boldsymbol{\theta}, \mathbf{x}, Rh(\mathbf{z}^\xi)) = \mathbb{P}\left( \bigwedge_{i=1}^{p} \left\{ \mathbf{k}_{i+p}^\mathsf{T} \mathbf{z}^\xi + b_{i+p} \leq \nu_i \leq -\mathbf{k}_i^\mathsf{T} \mathbf{z}^\xi - b_i \right\} \right)$$

$$= \prod_{i=1}^{p} \mathbb{P}\left( \mathbf{k}_{i+p}^\mathsf{T} \mathbf{z}^\xi + b_{i+p} \leq \nu_i \leq -\mathbf{k}_i^\mathsf{T} \mathbf{z}^\xi - b_i \right) \tag{8}$$

$$= \prod_{i=1}^{p} \left\{ \Phi\left( -\mathbf{k}_i^\mathsf{T} \mathbf{z}^\xi - b_i \right) - \Phi\left( \mathbf{k}_{i+p}^\mathsf{T} \mathbf{z}^\xi + b_{i+p} \right) \right\},$$

where $\mathbf{k}_j^\mathsf{T}$ is $j$-th row of $\mathbf{MB}$, and $b_j$ is $j$-th component of $\mathbf{MAx} - \mathbf{d}$.

Besides, we prove the convexity of $\ell\left( \mathbf{z}^\xi \right)$ as follows. First, it can be derived that:

$$\ell\left( \mathbf{z}^\xi \right) = -\log \mathbb{P}(\mathbf{Y} \in \mathcal{S} \mid \boldsymbol{\theta}, \mathbf{x}, Rh(\mathbf{z}^\xi))$$

$$= -\sum_{i=1}^{p} \log\left( \Phi\left( -\mathbf{k}_i^\mathsf{T} \mathbf{z}^\xi - b_i \right) - \Phi\left( \mathbf{k}_{i+p}^\mathsf{T} \mathbf{z}^\xi + b_{i+p} \right) \right), \tag{9}$$

Let $\ell_i\left( \mathbf{z}^\xi \right)$ denote $-\log\left( \Phi\left( -\mathbf{k}_i^\mathsf{T} \mathbf{z}^\xi - b_i \right) - \Phi\left( \mathbf{k}_{i+p}^\mathsf{T} \mathbf{z}^\xi + b_{i+p} \right) \right)$. If we could prove that $\ell_i\left( \mathbf{z}^\xi \right)$s are all convex *w.r.t.* $\mathbf{z}^\xi$, it suffices to prove $\ell\left( \mathbf{z}^\xi \right)$ is convex *w.r.t.* $\mathbf{z}^\xi$ because $\ell\left( \mathbf{z}^\xi \right) = \sum_{i=1}^{p} \ell_i\left( \mathbf{z}^\xi \right)$.

It can be observed that $-\mathbf{k}_i = \mathbf{k}_{i+p}$ always holds due to the definition of $\mathbf{M}$. Meanwhile, it has to hold that $b_{i+p} \leq -b_i$ because $\mathcal{S}(\mathbf{M}, \mathbf{d})$ is a canonical rectangle as defined in Def. 3.2. Without loss of generality, assume the $j$-th component of $\mathbf{k}_i$ is non-zero, then define the affine transformation:

$$a_i\left( \mathbf{z}^\xi \right) = \mathbf{z}^\xi + \frac{b_{i+p}}{k_{ij}} \mathbf{e}_j, \text{ where } \mathbf{e}_j \text{ is a zero vector except } j\text{-th component set to 1}.$$

Notice that by definition, it can be derived that:

$$\ell_i\left( a_i\left( \mathbf{z}^\xi \right) \right) = -\log\left( \Phi\left( -\mathbf{k}_i^\mathsf{T} a_i\left( \mathbf{z}^\xi \right) - b_i \right) - \Phi\left( -\mathbf{k}_i^\mathsf{T} a_i\left( \mathbf{z}^\xi \right) + b_{i+p} \right) \right)$$

$$= -\log\left( \Phi\left( -\mathbf{k}_i^\mathsf{T} \mathbf{z}^\xi - b_{i+p} - b_i \right) - \Phi\left( -\mathbf{k}_i^\mathsf{T} \mathbf{z}^\xi \right) \right).$$

Because $-b_{i+p} - b_i > 0$ as explained above, $\ell_i\left( a_i\left( \mathbf{z}^\xi \right) \right)$ is convex *w.r.t.* $\mathbf{z}^\xi$ by Lemma B.1. Since $a_i(\cdot)$ is an affine transformation, $\ell_i\left( \mathbf{z}^\xi \right)$ is also convex *w.r.t.* $\mathbf{z}^\xi$ (for $\forall i$), which suffices to prove that $\ell\left( \mathbf{z}^\xi \right)$ is convex *w.r.t.* $\mathbf{z}^\xi$.

$\square$

## B.2. Proof of Theorem 3.8

The proof of Thm. 3.8 relies on the detailed definition of the gradient, the Hessain matrix, the DiagApprox operator and the Armijo-like rule. Hence, we provide the proof in Appx. C after the details of Alg. 1.

## B.3. Proof of Proposition 3.10

**Proposition 3.10.** *Let $\mathcal{S}$ denote a non-empty region in $\mathbb{R}^d$, and let $\mathcal{I}_{cr}(\mathbf{M}, \mathbf{d})$ denote the inner CARE embedding of $\mathcal{S}$. Then, it always holds that $\ell(\mathbf{z}^\xi) \leq \hat{\ell}(\mathbf{z}^\xi)$, where the loss function $\ell(\mathbf{z}^\xi) = -\log\left( \mathbb{P}(\mathbf{Y} \in \mathcal{S} \mid \hat{\boldsymbol{\theta}}, \mathbf{x}, Rh(\mathbf{z}^\xi)) \right)$ and $\hat{\ell}(\mathbf{z}^\xi) = -\log\left( \mathbb{P}(\mathbf{Y} \in \mathcal{I}_{cr} \mid \hat{\boldsymbol{\theta}}, \mathbf{x}, Rh(\mathbf{z}^\xi)) \right)$.*

*Proof.* By definition of the inner CARE embedding as in Def. 3.9, it always holds that $\mathcal{I}_{cr}(\mathbf{M}, \mathbf{d}) \subseteq \mathcal{S}$. Hence, it suffices to prove that for any given $\hat{\boldsymbol{\theta}}$, $\mathbf{x}$, and $\mathbf{z}^\xi$, it must hold that $\mathbb{P}(\mathbf{Y} \in \mathcal{I}_{cr} \mid \hat{\boldsymbol{\theta}}, \mathbf{x}, Rh(\mathbf{z}^\xi)) \leq \mathbb{P}(\mathbf{Y} \in \mathcal{S} \mid \hat{\boldsymbol{\theta}}, \mathbf{x}, Rh(\mathbf{z}^\xi))$ by the definition of probability theory. Consequently, we always have $\ell(\mathbf{z}^\xi) \leq \hat{\ell}(\mathbf{z}^\xi)$. $\square$

## B.4. Proof of Proposition 3.11

**Proposition 3.11.** *Let $\mathcal{S} = \{\mathbf{x} \in \mathbb{R}^d \mid \|\mathbf{x} - \boldsymbol{o}\|_2^2 = r^2\}$ denote a circular region on $\mathbb{R}^d$, and let $\mathbf{C}, \boldsymbol{\Sigma}$ denote problem-related matrices as defined in Lemma 3.4. Then, the inner CARE embedding of $\mathcal{S}$ can be expressed by $\mathcal{I}_{cr}(\mathbf{M}, \mathbf{d})$ with:*

$$\begin{cases} \mathbf{M} = (\mathbf{I}, -\mathbf{I})^\mathsf{T} \boldsymbol{\Lambda} \mathbf{Q} \\ \mathbf{d} = (\mathbf{I}, \mathbf{I})^\mathsf{T} \boldsymbol{\Lambda} \mathbf{1} \cdot r/\sqrt{d} + \mathbf{Mo}, \end{cases}$$

*where $\mathbf{Q}^\mathsf{T} \boldsymbol{\Lambda} \mathbf{Q}$ is the eigen-decomposition of matrix $\mathbf{C}\boldsymbol{\Sigma}\mathbf{C}^\mathsf{T}$, $\mathbf{I} \in \mathbb{R}^{d \times d}$ is the identity matrix and $\mathbf{1} \in \mathbb{R}^d$ is the vector with all components equals 1.*

*Proof.* The inner CARE rectangle of a circle must be a (hyper-)square. Let $a$ denote the edge length of the square, then by pythagorean theorem, it can be derived that:

$$d \cdot a^2 = (2r)^2 \Rightarrow a = \frac{2r}{\sqrt{d}}.$$

Hence, the inner CARE square *w.r.t.* the original axes coordinate system should be:

$$\left\{ \mathbf{x} \mid (\mathbf{I}, -\mathbf{I})^\mathsf{T}(\mathbf{x} - \mathbf{o}) \le (\mathbf{1}, \mathbf{1})^\mathsf{T} \frac{r}{\sqrt{d}} \right\}.$$

If we rotate the coordinate system using the positive definite matrix $\mathbf{Q}$, the inner CARE square *w.r.t.* $\mathbf{Q}$ can be expressed as:

$$\left\{ \mathbf{x} \mid (\mathbf{I}, -\mathbf{I})^\mathsf{T}\mathbf{Q}(\mathbf{x} - \mathbf{o}) \le (\mathbf{1}, \mathbf{1})^\mathsf{T} \frac{r}{\sqrt{d}} \right\}.$$

As introduced in Sec. 3.1, we add the shrinkage matrix $\boldsymbol{\Lambda}$, and can derive that:

$$\left\{ \mathbf{x} \mid (\mathbf{I}, -\mathbf{I})^\mathsf{T}\boldsymbol{\Lambda}\mathbf{Q}(\mathbf{x} - \mathbf{o}) \le (\mathbf{I}, \mathbf{I})^\mathsf{T}\boldsymbol{\Lambda}\mathbf{1} \frac{r}{\sqrt{d}} \right\},$$

from which we can assert that $\mathbf{M} = (\mathbf{I}, -\mathbf{I})^\mathsf{T}\boldsymbol{\Lambda}\mathbf{Q}$ and $\mathbf{d} = (\mathbf{I}, \mathbf{I})^\mathsf{T}\boldsymbol{\Lambda}\mathbf{1} \cdot r/\sqrt{d} + \mathbf{Mo}$ for region $\{\mathbf{x} \mid \mathbf{Mx} \le \mathbf{d}\}$. $\qquad \square$

## B.5. Proof of Theorem 4.1

**Theorem 4.1.** *When $|\mathbf{Y}| = 1$, let $\mathbb{P}_A^\star$ denote the maximal AUF probability $\mathbb{P}(Y \in [l, r] \mid \hat{\boldsymbol{\theta}}, \mathbf{x}, Rh(\mathbf{z}^\xi))$ that can be achieved by alteration $\mathbf{z}^\xi \in [\mathbf{z}_{\text{left}}, \mathbf{z}_{\text{right}}]$, and let $\mathbf{z}^\star$ denote the alteration outputted by Alg. 2. Then the probability $\mathbb{P}(Y \in [l, r] \mid \hat{\boldsymbol{\theta}}, \mathbf{x}, Rh(\mathbf{z}^\star))$ equals the maximal one $\mathbb{P}_A^\star$.*

*Proof.* Recall from Sec. 4 that when the dimension of $\mathbf{Y}$ is 1, we can know that:

$$\ell(\mathbf{z}^\xi) = -\log\left\{ \Phi\left(\mathbf{k}_2^\mathsf{T}\mathbf{z}^\xi - b_1\right) - \Phi\left(\mathbf{k}_2^\mathsf{T}\mathbf{z}^\xi + b_2\right) \right\}.$$

In this case, to find $\mathbf{z}_\star^\xi \; \underset{\mathbf{z}^\xi \in [\mathbf{z}_{\text{left}}, \mathbf{z}_{\text{right}}]}{\arg\min} \; \ell(\mathbf{z}^\xi)$, we first define function $f(x)$ as follws:

$$f(x) = -\log\left\{ \Phi\left(x - b_1\right) - \Phi\left(x + b_2\right) \right\}.$$

Let $m = \mathbf{k}_2^\mathsf{T} \cdot (\mathbb{I}(\mathbf{k}_2 > 0) \circ \mathbf{z}_{\text{left}} + \mathbb{I}(\mathbf{k}_2 < 0) \circ \mathbf{z}_{\text{right}})$ and $M = \mathbf{k}_2^\mathsf{T} \cdot (\mathbb{I}(\mathbf{k}_2 < 0) \circ \mathbf{z}_{\text{left}} + \mathbb{I}(\mathbf{k}_2 > 0) \circ \mathbf{z}_{\text{right}})$, it can be derived that $\mathbf{k}_2^\mathsf{T}\mathbf{z}^\xi \in [m, M]$ and $\mathbf{k}_2^\mathsf{T}\mathbf{z}^\xi$ can continuously choose the value in this interval. Hence, we turn to use a two stage method to find $\mathbf{z}_\star^\xi$: first, we find $x_\star = \underset{x \in [m, M]}{\arg\min} f(x)$; and second, we find an appropriate $\mathbf{z}_\star^\xi$ such that $\mathbf{k}_2^\mathsf{T}\mathbf{z}_\star^\xi = x_\star$.

It can be derived that $f'(x) = -\frac{\phi(x - b_1) - \phi(x + b_2)}{\Phi(x - b_1) - \Phi(x + b_2)}$, which can be analyzed that $f'(x) \le 0$ on $(-\infty, \frac{b_1 - b_2}{2}]$ and $f'(x) \ge 0$ on $[\frac{b_1 - b_2}{2}, +\infty)$. Hence, $f(x)$ is monotone decreasing on $(-\infty, \frac{b_1 - b_2}{2}]$ and is monotone increasing on $[\frac{b_1 - b_2}{2}, +\infty)$. Thus, it can be derived that if $\frac{b_1 - b_2}{2} \le m$, then $x_\star = m$; and if $\frac{b_1 - b_2}{2} \ge M$, then $x_\star = M$. For these two cases, we can directly set

$\mathbf{z}_\star^\xi = \mathbb{I}(\mathbf{k}_2 > \mathbf{0}) \circ \mathbf{z}_{\text{left}} + \mathbb{I}(\mathbf{k}_2 < \mathbf{0}) \circ \mathbf{z}_{\text{right}}$ for $x_\star = m$ and set $\mathbf{z}_\star^\xi = \mathbb{I}(\mathbf{k}_2 < \mathbf{0}) \circ \mathbf{z}_{\text{left}} + \mathbb{I}(\mathbf{k}_2 > \mathbf{0}) \circ \mathbf{z}_{\text{right}}$ for $x_\star = M$ such that $\mathbf{k}_2^\mathsf{T} \mathbf{z}_\star^\xi = x_\star$ can always hold.

For the remaining case that $m < \frac{b_1 - b_2}{2} < M$, it holds that $x_\star = \frac{b_1 - b_2}{2}$. In this case, to find an appropriate $\mathbf{z}_\star^\xi$ such that $\mathbf{k}_2^\mathsf{T} \mathbf{z}_\star^\xi = \frac{b_1 - b_2}{2}$, we can start from $\mathbf{z}_\star^\xi = \mathbf{0}$ and perform an iteration that for each dimension $j$ of $\mathbf{z}_\star^\xi$, check if there exist $z_j \in [z_{\text{left},j}, z_{\text{right},j}]$ such that substitute the $j$-th dimension of $\mathbf{z}_\star^\xi$ by $z_j$ can lead to $\mathbf{k}_2^\mathsf{T} \mathbf{z}_\star^\xi = \frac{b_1 - b_2}{2}$. The checking process is listed in line 9 to line 14 in Alg. 2. As we have discussed how to determine $\mathbf{z}_\star^\xi$ for all the possible cases ($\frac{b_1 - b_2}{2} \leq m$, $m < \frac{b_1 - b_2}{2} < M$ and $\frac{b_1 - b_2}{2} \geq M$), it has been proven that Alg. 2 always outputs the alteration that can achieve $\min_{\mathbf{z}^\xi \in [\mathbf{z}_{\text{left}}, \mathbf{z}_{\text{right}}]} \ell(\mathbf{z}^\xi)$, *i.e.*, the one that can achieve the maximum AUF probability. $\qquad\square$

## C. Details of projection Newton method

We aim to optimize the following objective function:

$$\ell\left(\mathbf{z}^\xi\right) = \sum_{i=1}^d -\log\left\{\Phi\left(-\mathbf{k}_i^\mathsf{T}\mathbf{z}^\xi - b_i\right) - \Phi\left(-\mathbf{k}_i^\mathsf{T}\mathbf{z}^\xi + b_{i+d}\right)\right\}.$$

We can compute both the gradient and the second-order Hessian matrix of the objective function as follows:

$$
\begin{aligned}
\mathbf{g}\left(\mathbf{z}^\xi\right) &\triangleq \nabla\ell\left(\mathbf{z}^\xi\right) = \sum_{i=1}^d \frac{\phi\left(-\mathbf{k}_i^\mathsf{T}\mathbf{z}^\xi - b_i\right) - \phi\left(-\mathbf{k}_i^\mathsf{T}\mathbf{z}^\xi + b_{i+d}\right)}{\Phi\left(-\mathbf{k}_i^\mathsf{T}\mathbf{z}^\xi - b_i\right) - \Phi\left(-\mathbf{k}_i^\mathsf{T}\mathbf{z}^\xi + b_{i+d}\right)}\mathbf{k}_i \\
\mathbf{H}\left(\mathbf{z}^\xi\right) &\triangleq \nabla^2\ell\left(\mathbf{z}^\xi\right) = \sum_{i=1}^d \mathbf{k}_i^\mathsf{T}\mathbf{k}_i \cdot \left\{ \frac{\left(\phi\left(-\mathbf{k}_i^\mathsf{T}\mathbf{z}^\xi - b_i\right) - \phi\left(-\mathbf{k}_i^\mathsf{T}\mathbf{z}^\xi + b_{i+d}\right)\right)^2}{\left(\Phi\left(-\mathbf{k}_i^\mathsf{T}\mathbf{z}^\xi - b_i\right) - \Phi\left(-\mathbf{k}_i^\mathsf{T}\mathbf{z}^\xi + b_{i+d}\right)\right)^2} + \right. \\
&\qquad \left. \frac{\left\{\left(-\mathbf{k}_i^\mathsf{T}\mathbf{z}^\xi - b_i\right)\phi\left(-\mathbf{k}_i^\mathsf{T}\mathbf{z}^\xi - b_i\right) - \left(-\mathbf{k}_i^\mathsf{T}\mathbf{z}^\xi + b_{i+d}\right)\phi\left(-\mathbf{k}_i^\mathsf{T}\mathbf{z}^\xi + b_{i+d}\right)\right\}}{\Phi\left(-\mathbf{k}_i^\mathsf{T}\mathbf{z}^\xi - b_i\right) - \Phi\left(-\mathbf{k}_i^\mathsf{T}\mathbf{z}^\xi + b_{i+d}\right)} \right\}
\end{aligned}
\tag{10}
$$

Before proceeding, we make a few simple observations regarding $\mathbf{g}(\mathbf{z}^\xi)$ and $\mathbf{H}(\mathbf{z}^\xi)$. Both of these quantities admit closed-form expressions, enabling us to leverage second-order Hessian information to facilitate the optimization of $\ell(\mathbf{z}^\xi)$, and we employ a modified projection Newton method. The method can be expressed as follows:

$$\mathbf{z}_{k+1} = \mathbf{z}_k(\alpha_k),$$

where the subscript $k + 1$ and $k$ are identical to $(t + 1)$ and $(t)$ as defined in the main body of the paper, and

$$\mathbf{z}_k(\alpha) = \text{Proj}\left[\mathbf{z}_k - \alpha\mathbf{D}_k\nabla\ell\left(\mathbf{z}_k\right)\right].$$

For all $\mathbf{z} \in \mathbb{R}^n$, we define the projection operator $\text{Proj}[\mathbf{z}]$ with coordinates given by:

$$\text{Proj}[\mathbf{z}]^i = \begin{cases} \mathbf{z}_{\text{right}}^i & \text{if } \mathbf{z}_{\text{right}}^i \leq \mathbf{z}^i, \\ \mathbf{z}^i & \text{if } \mathbf{z}_{\text{left}}^i < \mathbf{z}^i < \mathbf{z}_{\text{right}}^i, \\ \mathbf{z}_{\text{left}}^i & \text{if } \mathbf{z}^i \leq \mathbf{z}_{\text{left}}^i. \end{cases}$$

To clarify the definitions of $\mathbf{D}_k$ and $\alpha_k$ in the above equation, we first introduce the subset of indices $\mathcal{I}_k^{\text{Proj}}$:

$$\mathcal{I}_k^{\text{Proj}} = \left\{i \mid \mathbf{z}_{\text{left}}^i \leq \mathbf{z}_k^i \leq \mathbf{z}_{\text{left}}^i + \varepsilon_k \text{ and } \frac{\partial\ell\left(\mathbf{z}_k\right)}{\partial z^i} > 0 \text{ or } \mathbf{z}_{\text{right}}^i - \varepsilon_k \leq \mathbf{z}_k^i \leq \mathbf{z}_{\text{right}}^i \text{ and } \frac{\partial\ell\left(\mathbf{z}_k\right)}{\partial\mathbf{z}^i} < 0\right\},$$

and the matrix $\mathbf{D}_k$ is defined as:

$$\mathbf{D}_k = \mathbf{H}_k^{-1},$$

where $\mathbf{H}_k$ is the matrix with elements $\mathbf{H}_k^{ij}$ given by

$$\mathbf{H}_k^{ij} = \begin{cases} 0 & \text{if } i \neq j, \text{ and either } i \in \mathcal{I}_k^{\text{Proj}} \text{ or } j \in \mathcal{I}_k^+, \\ \frac{\partial^2 \ell(x_k)}{\partial \mathbf{z}^i \partial \mathbf{z}^i} & \text{otherwise.} \end{cases}$$

The algorithm utilizes a scalar $\varepsilon > 0$ (typically small), a fixed diagonal positive definite matrix $M$ (such as the identity matrix), and two parameters $\beta \in (0,1)$ and $\sigma \in \left(0, \frac{1}{2}\right)$, which are used in conjunction with an Armijo-like stepsize rule (first introduced in (Bertsekas, 1976)). The scalar $\varepsilon_k$ is defined as

$$\varepsilon_k = \min \left\{ \varepsilon, \left| \mathbf{z}_k - [\mathbf{z}_k - \mathbf{M}\nabla\ell(\mathbf{z}_k)]^* \right| \right\}.$$

The matrix $\mathbf{D}_k$ is positive definite, and $\mathbf{M}$ is a fixed diagonal positive definite matrix. The stepsize $\alpha_k$ is given by:

$$\alpha_k = \beta^{m_k},$$

where

$$p_k = \mathbf{D}_k \nabla\ell(x_k),$$

and $m_k$ is the first nonnegative integer $m$ such that:

$$\ell(\mathbf{z}_k) - \ell[\mathbf{z}_k(\beta^m)] \geq \sigma \left\{ \beta^m \sum_{i \in I_k^{\text{Proj}}} \frac{\partial \ell(\mathbf{z}_k)}{\partial \mathbf{z}^i} p_k^i + \sum_{i \in I_k^{\text{Proj}}} \frac{\partial \ell(\mathbf{z}_k)}{\partial \mathbf{z}^i} \left[ \mathbf{z}_k^i - \mathbf{z}_k^i(\beta^m) \right] \right\}. \tag{11}$$

**Proposition C.1.** *We have the following observations:*

1 *The gradient $\nabla\ell$ is Lipschitz continuous on each bounded set of $\mathbb{R}^n$; i.e., given any bounded set $S \subset R^n$ there exists a scalar $L$ (depending on $S$ ) such that*

$$|\nabla\ell(\mathbf{z}_1) - \nabla\ell(\mathbf{z}_2)| \leq L|\mathbf{z}_1 - \mathbf{z}_2| \quad \forall \mathbf{z}_1, \mathbf{z}_2 \in S. \tag{12}$$

2 *There exist positive scalars $\lambda_1, \lambda_2$ and nonnegative integers $q_1, q_2$ such that*

$$\lambda_1 w_k^{q_1} |\mathbf{z}|^2 \leq \mathbf{z}' \mathbf{D}_k \mathbf{z} \leq \lambda_2 w_k^{q_2} |\mathbf{z}|^2 \quad \forall \mathbf{z} \in R^n, \quad k = 0, 1 \cdots, \tag{13}$$

*where*

$$w_k = \left| \mathbf{z}_k - [\mathbf{z}_k - M\nabla\ell(\mathbf{z}_k)]^{\text{Proj}} \right|.$$

3 *The local minimum $\mathbf{z}^*$ of problem is such that for some $\delta > 0$, $\ell$ is twice continuously differentiable in the open sphere $\{\mathbf{z} \mid |\mathbf{z} - \mathbf{z}^*| < \delta\}$, and there exist positive scalars $m_1, m_2$ such that*

$$m_1|\mathbf{z}|^2 \leq \mathbf{z}'\nabla^2\ell(\mathbf{z})\mathbf{z} \leq m_2|\mathbf{z}|^2 \quad \forall \mathbf{z} \text{ such that } |\mathbf{z} - \mathbf{z}^*| < \delta \text{ and } \mathbf{z} \neq 0 \text{ such that } \mathbf{z}^i = 0, \forall i \in B(\mathbf{z}^*).$$

*Furthermore,*

$$\frac{\partial \ell(\mathbf{z}^*)}{\partial \mathbf{z}^i} > 0 \quad \forall i \in B(\mathbf{z}^*). \tag{14}$$

*Proof.* Consider a bounded set $\mathcal{Z} \subset \mathbb{R}^d$ for $\mathbf{z}^\xi$. Because $\mathbf{k}_i$, $b_i$, and $b_{i+d}$ are fixed, the quantities $-\mathbf{k}_i^\top \mathbf{z}^\xi - b_i$ and $-\mathbf{k}_i^\top \mathbf{z}^\xi + b_{i+d}$ vary over a bounded interval as $\mathbf{z}^\xi$ ranges over the bounded set $Z$.

Since $\Phi(\cdot)$ is continuous, $\Phi\left(-\mathbf{k}_i^\top \mathbf{z}^\xi - b_i\right)$ and $\Phi\left(-\mathbf{k}_i^\top \mathbf{z}^\xi + b_{i+d}\right)$ are also continuous functions of $\mathbf{z}^\xi$, and hence uniformly continuous on the compact set $\mathcal{Z}$.

We write $b_{\min} = \min_{1 \le i \le d} |b_{i+d} - b_i|$, and let

$$D_i\left(\mathbf{z}^\xi\right) = \Phi\left(-\mathbf{k}_i^\top \mathbf{z}^\xi + b_{i+d}\right) - \Phi\left(-\mathbf{k}_i^\top \mathbf{z}^\xi - b_i\right). \tag{15}$$

Without loss of generality, suppose $b_{i+d} > b_i$. Then for all $\mathbf{z}^\xi \in \mathcal{Z}$:

$$-\mathbf{k}_i^\top \mathbf{z}^\xi - b_i < -\mathbf{k}_i^\top \mathbf{z}^\xi + b_{i+d}.$$

Since $\Phi(\cdot)$ is strictly increasing, it follows that $D_i\left(\mathbf{z}^\xi\right) > 0$. Because $\mathbf{z}^\xi$ is restricted to a bounded set $\mathcal{Z}$, the values $-\mathbf{k}_i^\top \mathbf{z}^\xi$ lie within some finite interval. Hence, $\left(-\mathbf{k}_i^\top \mathbf{z}^\xi - b_i\right)$ and $\left(-\mathbf{k}_i^\top \mathbf{z}^\xi + b_{i+d}\right)$ also lie in some bounded intervals. Since $\Phi(\cdot)$ is continuous and strictly increasing, the difference $\Phi\left(-\mathbf{k}_i^\top \mathbf{z}^\xi + b_{i+d}\right) - \Phi\left(-\mathbf{k}_i^\top \mathbf{z}^\xi - b_i\right)$ attains a strictly positive minimum on the compact set $\mathcal{Z}$. Alternatively, one can use a Lipschitz continuity argument: $\Phi(\cdot)$ is Lipschitz continuous with some constant $L \le \frac{1}{\sqrt{2\pi}}$, so

$$D_i\left(\mathbf{z}^\xi\right) = \Phi(y) - \Phi(x) \ge L|y - x|,$$

where $x = -\mathbf{k}_i^\top \mathbf{z}^\xi - b_i$ and $y = -\mathbf{k}_i^\top \mathbf{z}^\xi + b_{i+d}$, and $L$ is determined by $S$. Thus,

$$|y - x| = |b_{i+d} - b_i| \ge b_{\min}.$$

Therefore,

$$D_i\left(\mathbf{z}^\xi\right) \ge L b_{\min} \quad \forall \mathbf{z}^\xi \in Z$$

For the numerator:

$$N_i\left(\mathbf{z}^\xi\right) = \phi\left(-\mathbf{k}_i^\top \mathbf{z}^\xi - b_i\right) - \phi\left(-\mathbf{k}_i^\top \mathbf{z}^\xi + b_{i+d}\right).$$

Since $\phi(\cdot)$ is the standard normal PDF, it is bounded above by $\frac{1}{\sqrt{2\pi}}$. Thus:

$$\left|N_i\left(\mathbf{z}^\xi\right)\right| \le \frac{2}{\sqrt{2\pi}}.$$

Combining these results:

$$\left|\frac{N_i\left(\mathbf{z}^\xi\right)}{D_i\left(\mathbf{z}^\xi\right)}\right| \le \frac{\frac{2}{\sqrt{2\pi}}}{L b_{\min}} = \frac{C_1}{b_{\min}},$$

where $C_1 = \frac{2}{\sqrt{2\pi} L}$ is a constant independent of $\mathbf{z}^\xi$. Since $\mathbf{k}_i$ are fixed vectors, let

$$C_2 = \max_{1 \le i \le d} \|\mathbf{k}_i\|,$$

then each term in the sum has a uniform bound:

$$\left\| \frac{N_i\left(\mathbf{z}^\xi\right)}{D_i\left(\mathbf{z}^\xi\right)} \mathbf{k}_i \right\| \leq \frac{C_1}{b_{\min}} C_2.$$

Summing over $i = 1, \ldots, d$:

$$\|\mathbf{g}\left(\mathbf{z}^\xi\right)\| = \left\| \sum_{i=1}^d \frac{N_i\left(\mathbf{z}^\xi\right)}{D_i\left(\mathbf{z}^\xi\right)} \mathbf{k}_i \right\| \leq \sum_{i=1}^d \left\| \frac{N_i\left(\mathbf{z}^\xi\right)}{D_i\left(\mathbf{z}^\xi\right)} \mathbf{k}_i \right\| \leq \sum_{i=1}^d \frac{C_1}{b_{\min}} C_2 = d \frac{C_1 C_2}{b_{\min}}.$$

This constants do not depend on $\mathbf{z}^\xi$.

For the second one, From this definition, $\mathbf{H}_k$ is essentially diagonal except possibly for som modifications dictated by the sets $\mathcal{I}_k^{\mathrm{Proj}}$ and $\mathcal{I}_k^+$. Crucially, if an index $i$ falls into $\mathcal{I}_k^{\mathrm{Proj}}$, it indicates that $\mathbf{z}_k^i$ is close to the boundary and that the sign of $\frac{\partial \ell(\mathbf{z}_k)}{\partial z^i}$ is such that pushing beyond the boundary would reduce, hence the zeroing out of certain off-diagonal terms ensures a stable modification.

Because $\ell(\cdot)$ is twice continuously differentiable, the second derivatives $\frac{\partial^2 \ell}{\partial z^i \partial z^i}$ are continuous and thus bounded within any compact region. This ensures that each nonzero diagonal element of $\mathbf{H}_k$ stays within certain positive bounds, except potentially when $i \in I_k^{\mathrm{Proj}}$, where the proximity to the boundary and the definition of $\mathbf{H}_k$ allows the diagonal entries to shrink.

Indices in the set $I_k^{\mathrm{Proj}}$ correspond to coordinates near the boundary where the step is adjusted by the projection. The deviation $w_k$ controls how significant this projection adjustment is. As $\mathbf{z}_k$ approaches the boundary, the difference between $\mathbf{z}_k$ and $[\mathbf{z}_k - M\nabla \ell\left(\mathbf{z}_k\right)]^{\mathrm{Proj}}$ reflects how the Hessian's diagonal elements may shrink or grow.

By suitable scaling arguments and using continuity of the second derivatives, we can show that there exist nonnegative integers $q_1, q_2$ and positive constants $\lambda_1, \lambda_2$ such that every nonzero diagonal element of $\mathbf{H}_k$ (and thus its inverse $\mathbf{D}_k$) can be sandwiched as follows:

$$\lambda_1 w_k^{q_1} \leq \mathbf{H}_k^{ii} \leq \lambda_2 w_k^{q_2}.$$

Since $\mathbf{D}_k = \mathbf{H}_k^{-1}$, these inequalities invert to give:

$$\frac{1}{\lambda_2} w_k^{-q_2} \leq \frac{1}{\mathbf{H}_k^{ii}} \leq \frac{1}{\lambda_1} w_k^{-q_1}.$$

Taking all coordinates $\mathbf{z}$ into account and using the fact that $\mathbf{D}_k$ is diagonal (due to the structure imposed by zeroing out certain offdiagonal terms), for all $\mathbf{z} \in \mathbb{R}^n$, we obtain:

$$\lambda_1 w_k^{q_1} |\mathbf{z}|^2 \leq \mathbf{z}' \mathbf{D}_k \mathbf{z} \leq \lambda_2 w_k^{q_2} |\mathbf{z}|^2.$$

For the third one, $\ell(\cdot)$ is twice continuously differentiable at $\mathbf{z}^*$. By definition, this means that there exists a $\delta > 0$ such that $\ell$ is $C^2$-smooth in the open ball

$$\{\mathbf{z} \mid \|\mathbf{z} - \mathbf{z}^*\| < \delta\}$$

This ensures that both the gradient $\nabla \ell(\mathbf{z})$ and the Hessian $\nabla^2 \ell(\mathbf{z})$ are continuous functions in that neighborhood. Since $\mathbf{z}^*$ is a local minimizer of $\ell(\cdot)$, the second-order sufficient condition for a strict local minimum states that the Hessian at $\mathbf{z}^*$, *i.e.*, $\nabla^2 \ell\left(\mathbf{z}^*\right)$, is positive definite.

Because $\nabla^2 \ell\left(\mathbf{z}^*\right)$ is a symmetric positive definite matrix, it has a full set of positive eigenvalues. Let $\lambda_{\min}\left(\mathbf{z}^*\right)$ and $\lambda_{\max}\left(\mathbf{z}^*\right)$ denote the smallest and largest eigenvalues of $\nabla^2 \ell\left(\mathbf{z}^*\right)$, respectively. We then have:

$$0 < \lambda_{\min}\left(\mathbf{z}^*\right) \leq \frac{\mathbf{h}^\top \nabla^2 \ell\left(\mathbf{z}^*\right) \mathbf{h}}{\|\mathbf{h}\|^2} \leq \lambda_{\max}\left(\mathbf{z}^*\right)$$

And the Hessian $\nabla^2 \ell(\mathbf{z})$ is continuous in $\mathbf{z}$. Therefore, there exists a $\delta > 0$ such that for all $\mathbf{z}$ with $\|\mathbf{z} - \mathbf{z}^*\| < \delta$, the Hessian $\nabla^2 \ell(\mathbf{z})$ remains close to $\nabla^2 \ell\left(\mathbf{z}^*\right)$ in operator norm. This implies that the eigenvalues of $\nabla^2 \ell(\mathbf{z})$ are also close to those of $\nabla^2 \ell\left(\mathbf{z}^*\right)$.

Since $\nabla^2 \ell\left(\mathbf{z}^*\right)$ is positive definite, we can find constants $m_1, m_2 > 0$ such that for all $\mathbf{z}$ with $\|\mathbf{z} - \mathbf{z}^*\| < \delta$ :

$$m_1 \mathbf{I} \preceq \nabla^2 \ell(\mathbf{z}) \preceq m_2 \mathbf{I},$$

where $\mathbf{I}$ is the identity matrix and $\preceq$ denotes the Loewner (matrix) order. Equivalently:

$$m_1 |\mathbf{z}|^2 \leq \mathbf{z}^\top \nabla^2 \ell(\mathbf{z})\mathbf{z} \leq m_2 |\mathbf{z}|^2 \quad \forall \mathbf{z} \neq 0 \text{ with } \|\mathbf{z} - \mathbf{z}^*\| < \delta.$$

This shows both the boundedness from above and below of the quadratic form defined by the Hessian in a neighborhood of the minimizer. $\frac{\partial \ell(\mathbf{z}^*)}{\partial \mathbf{z}^i} > 0 \quad \forall i \in B\left(\mathbf{z}^*\right)$ is obviously since $z^*$ is local minimum and $\frac{\partial \ell(\mathbf{z}^*)}{\partial \mathbf{z}^i}$ is continous.

$\square$

Notice that in:

$$\mathbf{H}\left(\mathbf{z}^\xi\right) = \sum_{i=1}^d \mathbf{k}_i^\top \mathbf{k}_i \cdot \left\{ \frac{\left(\phi\left(-\mathbf{k}_i^\top \mathbf{z}^\xi - b_i\right) - \phi\left(-\mathbf{k}_i^\top \mathbf{z}^\xi + b_{i+d}\right)\right)^2}{\left(\Phi\left(-\mathbf{k}_i^\top \mathbf{z}^\xi - b_i\right) - \Phi\left(-\mathbf{k}_i^\top \mathbf{z}^\xi + b_{i+d}\right)\right)^2} + \right.$$
$$\left. \frac{\left\{\left(-\mathbf{k}_i^\top \mathbf{z}^\xi - b_i\right) \phi\left(-\mathbf{k}_i^\top \mathbf{z}^\xi - b_i\right) - \left(-\mathbf{k}_i^\top \mathbf{z}^\xi + b_{i+d}\right) \phi\left(-\mathbf{k}_i^\top \mathbf{z}^\xi + b_{i+d}\right)\right\}}{\Phi\left(-\mathbf{k}_i^\top \mathbf{z}^\xi - b_i\right) - \Phi\left(-\mathbf{k}_i^\top \mathbf{z}^\xi + b_{i+d}\right)} \right\},$$

since $\mathbf{k}_j^\top \mathbf{k}_j$ has only rank 1, the upper and lower bounds of the eigenvalues of $\mathbf{H}\left(\mathbf{z}^\xi\right)$ can be only depended on term:

$$\frac{\left(\phi\left(-\mathbf{k}_i^\top \mathbf{z}^\xi - b_i\right) - \phi\left(-\mathbf{k}_i^\top \mathbf{z}^\xi + b_{i+d}\right)\right)^2}{\left(\Phi\left(-\mathbf{k}_i^\top \mathbf{z}^\xi - b_i\right) - \Phi\left(-\mathbf{k}_i^\top \mathbf{z}^\xi + b_{i+d}\right)\right)^2} + \frac{\left(-\mathbf{k}_i^\top \mathbf{z}^\xi - b_i\right) \phi\left(-\mathbf{k}_i^\top \mathbf{z}^\xi - b_i\right) - \left(-\mathbf{k}_i^\top \mathbf{z}^\xi + b_{i+d}\right) \phi\left(-\mathbf{k}_i^\top \mathbf{z}^\xi + b_{i+d}\right)}{\Phi\left(-\mathbf{k}_i^\top \mathbf{z}^\xi - b_i\right) - \Phi\left(-\mathbf{k}_i^\top \mathbf{z}^\xi + b_{i+d}\right)}.$$

Next, we will prove that every limit point of a sequence $\{\mathbf{z}_k\}$ generated by iteration is a critical point with respect to Eq. (3). First, we need the following lemma, which was first proven in (Bertsekas, 1982).

**Lemma C.2.** *(Bertsekas, 1982).*

*Let $\mathbf{z} \geq 0$ and $D$ be a positive definite symmetric matrix that is diagonal with respect to $\mathcal{I}^{\text{Proj}}(\mathbf{z})$. Define*

$$\mathbf{z}(\alpha) = \text{Proj}[\mathbf{z} - \alpha \mathbf{D} \nabla \ell(\mathbf{z})] \quad \forall \alpha \geq 0.$$

*1 The vector $\mathbf{z}$ is a critical point with respect to problem (1) if and only if*

$$\mathbf{z} = \mathbf{z}(\alpha) \quad \forall \alpha \geq 0.$$

*2 If $\mathbf{z}$ is not a critical point with respect to problem (1), there exists a scalar $\bar{\alpha} > 0$ such that*

$$\ell[\mathbf{z}(\alpha)] < \ell(\mathbf{z}) \quad \forall \alpha \in (0, \bar{\alpha}].$$

Note that for all $k$ we have

$$\mathcal{I}_k^{\mathrm{Proj}} \supset \mathcal{I}^{\mathrm{Proj}}(\mathbf{z}_k),$$

so the matrix $\mathbf{D}_k$ is diagonal with respect to $I^{\mathrm{Proj}}(\mathbf{z}_k)$. It can be shown that for all $m \geq 0$, the right-hand side of Eq. (11) is nonnegative and is positive if and only if $\mathbf{z}_k$ is not a critical point. Indeed, since $\mathbf{D}_k$ is positive definite and diagonal with respect to $I_k^{\mathrm{Proj}}$, we have

$$\sum_{i \in r_k} \frac{\partial f(\mathbf{z}_k)}{\partial \mathbf{z}^i} p_k^i \geq 0 \quad \forall k = 0, 1, \cdots, \tag{16}$$

while for all $i \in I_k^{\mathrm{Proj}}$, given that $\frac{\partial \ell(\mathbf{z}^k)}{\partial \mathbf{z}^i} > 0$, we have $p_k^i > 0$ and thus

$$\begin{aligned}
\mathbf{z}_k^i - \mathbf{z}_k^i(\alpha) &\geq 0 \quad \forall \alpha \geq 0, \quad i \in I_k^{\mathrm{Proj}}, \quad k = 0, 1, \cdots, \\
\frac{\partial \ell(\mathbf{z}_k)}{\partial \mathbf{z}^i} \left[ \mathbf{z}_k^i - \mathbf{z}_k^i(\alpha) \right] &\geq 0 \quad \forall \alpha \geq 0, \quad i \in I_k^{\mathrm{Proj}}, \quad k = 0, 1, \cdots.
\end{aligned} \tag{17}$$

This shows that the right side of Eq. (11) is nonnegative. If $\mathbf{z}_k$ is not a critical point, then it is clear that one of the inequalities Eq. (16)is strict for $\alpha > 0$, making the right side of Eq. (11) positive for all $m \geq 0$. A slight modification of the proof of Proposition 1(b) also shows that if $\mathbf{z}_k$ is not a critical point, then Eq. (11) will be satisfied for all sufficiently large $m$, ensuring that the stepsize $\alpha_k$ is well-defined and can be determined via a finite number of arithmetic operations. If $\mathbf{z}_k$ is a critical point, then, by Lemma C.2, we have $\mathbf{z}_k = \mathbf{z}_k(\alpha)$ for all $\alpha \geq 0$. Furthermore, the argument given in the proof of Proposition 1(a) shows that:

$$\sum_{i=1}^{r} \frac{\partial \ell(\mathbf{z}_k)}{\partial \mathbf{z}^i} p_k^i = 0,$$

so both terms on the right side of Eq. (11) are zero. Since also $\mathbf{z}_k = \mathbf{z}_k(\alpha)$ for all $\alpha \geq 0$, it follows that Eq. (11) is satisfied for $m = 0$, thereby implying that:

$$\mathbf{z}_{k+1} = \mathbf{z}_k(1) = \mathbf{z}_k \quad \text{if } \mathbf{z}_k \text{ is critical.}$$

Now assume that there exists a subsequence $\{\mathbf{z}_k\}_k$ converging to a vector $\bar{\mathbf{z}}$ which is not critical. Since $\{\ell(\mathbf{z}_k)\}$ is decreasing and $f$ is continuous, it follows that $\{\ell(\mathbf{z}_k)\}$ converges to $f(\bar{\mathbf{z}})$ and therefore:

$$[\ell(\mathbf{z}_k) - \ell(\mathbf{z}_{k+1})] \to 0.$$

Since each of the sums on the right-hand side of Eq. (11) is nonnegative (cf. Eq. (16), Eq. (17)), we must have

$$\alpha_k \sum_{i \in I_k^{\mathrm{Proj}}} \frac{\partial \ell(\mathbf{z}_k)}{\partial \mathbf{z}^i} p_k^i \to 0; \tag{18}$$

and

$$\sum_{i \in r_k} \frac{\partial \ell(\mathbf{z}_k)}{\partial \mathbf{z}^i} \left[ \mathbf{z}_k^i - \mathbf{z}_k^i(\alpha_k) \right] \to 0.$$

Also, since $\bar{\mathbf{z}}$ is not critical and $M$ is diagonal, we have $\left| \bar{\mathbf{z}} - [\bar{\mathbf{z}} - M\nabla\ell(\bar{\mathbf{z}})]^{\mathrm{Proj}} \right| \neq 0$, so Eq. (13) implies that the eigenvalues of $\{\mathbf{D}_k\}_K$ are uniformly bounded above and away from zero. In view of the fact that $\mathbf{D}_k$ is diagonal *w.r.t.* $\mathcal{I}_k^{\mathrm{Proj}}$, it follows that there exist positive scalars $\bar{\lambda}_1, \bar{\lambda}_2$ such that for all $k \in K$ that are sufficiently large, it holds that:

$$\begin{aligned}
0 < \bar{\lambda}_1 \frac{\partial \ell(\mathbf{z}_k)}{\partial \mathbf{z}^i} &\leq p_k^i \leq \bar{\lambda}_2 \frac{\partial \ell(\mathbf{z}_k)}{\partial \mathbf{z}^i} \quad \forall i \in \mathcal{I}_k^{\mathrm{Proj}}, \\
\bar{\lambda}_1 \sum_{i \in \mathcal{I}_k} \left| \frac{\partial \ell(\mathbf{z}_k)}{\partial \mathbf{z}^i} \right|^2 &\leq \sum_{i \in \Gamma_k} p_k^i \frac{\partial \ell(\mathbf{z}_k)}{\partial \mathbf{z}^i} \leq \bar{\lambda}_2 \sum_{i \in \mathcal{I}_k} \left| \frac{\partial l\ell(\mathbf{z}_k)}{\partial \mathbf{z}^i} \right|^2.
\end{aligned}$$

We will show that our hypotheses so far lead to the conclusion that

$$\liminf_{\substack{k \to \infty \\ k \in K}} \alpha_k = 0. \tag{19}$$

Indeed, since $\bar{\mathbf{z}}$ is not a critical point, there must exist an index $i$ such that either

$$\bar{\mathbf{z}}^i > 0 \quad \text{and} \quad \frac{\partial f(\bar{\mathbf{z}})}{\partial \mathbf{z}^i} \neq 0, \tag{20}$$

or

$$\bar{\mathbf{z}}^i = 0 \quad \text{and} \quad \frac{\partial f(\bar{\mathbf{z}})}{\partial \mathbf{z}^i} < 0. \tag{21}$$

If $i \notin \mathcal{I}_k^{\mathrm{Proj}}$ for an infinite number of indices $k \in K$, then Eq. (19) follows from Eq. (18),Eq. (20) and Eq. (21) . If $i \in \mathcal{I}_k^{\mathrm{Proj}}$ for an infinite number of indices $k \in K$, then for all those indices we must have $\frac{\partial \ell(\mathbf{z}_k)}{\partial \mathbf{z}^i} > 0$, so Eq. (21) cannot hold. Therefore, from Eq. (20),

$$\bar{\mathbf{z}}^i > 0 \text{ and } \frac{\partial \ell(\bar{\mathbf{z}})}{\partial \mathbf{z}^i} > 0. \tag{22}$$

Since we have [cf. Eq. (17)] for all $k \in K$ for which $i \in \mathcal{I}_k^{\mathrm{Proj}}$, it holds that:

$$\sum_{j \in \mathcal{I}_k} \frac{\partial \ell(\mathbf{z}_k)}{\partial \mathbf{z}^i} \left[ \mathbf{z}_k^j - \mathbf{z}_k^j(\alpha_k) \right] \geq \frac{\partial \ell(\mathbf{z}_k)}{\partial \mathbf{z}^i} \left[ \mathbf{z}_k^i - \mathbf{z}_k^i(\alpha_k) \right] \geq 0,$$

which follows from Eq. (18) and Eq. (22) that

$$\lim_{\substack{k \to \infty \\ k \in K}} \left[ \mathbf{z}_k^i - \mathbf{z}_k^i(\alpha_k) \right] = 0.$$

Using the above relation, we obtain Eq. (19).

We will complete the proof by showing that $\{\alpha_k\}_K$ is bounded away from zero, thereby the subsequences $\{\mathbf{z}_k\}_K$, $\{p_k\}_K$, and $\{\mathbf{z}_k(\alpha)\}_K$, $\alpha \in [0,1]$, are uniformly bounded. Hence, by 12, there exists a scalar $L > 0$ such that for all $t \in [0,1]$, $\alpha \in [0,1]$, and $k \in K$, we have

$$|\nabla \ell(\mathbf{z}_k) - \nabla \ell[\mathbf{z}_k - t[\mathbf{z}_k - \mathbf{z}_k(\alpha)]]| \leq tL|\mathbf{z}_k - \mathbf{z}_k(\alpha)|.$$

For all $k \in K$ and $\alpha \in [0,1]$, we have

$$\ell[\mathbf{z}_k(\alpha)] = \ell(\mathbf{z}_k) + \nabla \ell(\mathbf{z}_k)'[\mathbf{z}_k(\alpha) - \mathbf{z}_k] + \int_0^1 \{\nabla \ell(\mathbf{z}_k) - \nabla \ell[\mathbf{z}_k - t[\mathbf{z}_k - \mathbf{z}_k(\alpha)]]\}' dt[\mathbf{z}_k - \mathbf{z}_k(\alpha)].$$

Hence,

$$\ell(\mathbf{z}_k) - \ell[\mathbf{z}_k(\alpha)] = \nabla \ell(\mathbf{z}_k)'[\mathbf{z}_k - \mathbf{z}_k(\alpha)] + \int_0^1 \{\nabla \ell[\mathbf{z}_k - t[\mathbf{z}_k - \mathbf{z}_k(\alpha)]] - \nabla \ell(\mathbf{z}_k)\}' dt[\mathbf{z}_k - \mathbf{z}_k(\alpha)]$$

$$\geq \nabla \ell(\mathbf{z}_k)'[\mathbf{z}_k - \mathbf{z}_k(\alpha)] - \int_0^1 |\nabla \ell[\mathbf{z}_k - t[\mathbf{z}_k - \mathbf{z}_k(\alpha)]] - \nabla \ell(\mathbf{z}_k)| dt |\mathbf{z}_k - \mathbf{z}_k(\alpha)|$$

$$\geq \nabla \ell(\mathbf{z}_k)'[\mathbf{z}_k - \mathbf{z}_k(\alpha)] - \frac{L}{2}|\mathbf{z}_k - \mathbf{z}_k(\alpha)|^2.$$

For $i \in \mathcal{I}_k^{\mathrm{Proj}}$, we have $\mathbf{z}_k^i(\alpha) = \left[\mathbf{z}_k^i - \alpha p_k^i\right]^{\mathrm{Proj}} \geq \mathbf{z}_k^i - \alpha p_k^i$ and $p_k^i > 0$, so $0 \leq \mathbf{z}_k^i - \mathbf{z}_k^i(\alpha) \leq \alpha p_k^i$. It follows,

$$\sum_{i \in r_k} \left| \mathbf{z}_k^i - \mathbf{z}_k^i(\alpha) \right|^2 \leq \alpha \sum_{i \in r_k} p_k^i \left[ \mathbf{z}_k^i - \mathbf{z}_k^i(\alpha) \right] \leq \alpha \tilde{\lambda}_2 \sum_{i \in r_k'} \frac{\partial \ell(\mathbf{z}_k)}{\partial \mathbf{z}^i} \left[ \mathbf{z}_k^i - \mathbf{z}_k^i(\alpha) \right].$$

Consider the sets

$$\mathcal{I}_{1,k} = \left\{ i \mid \frac{\partial \ell\left(\mathbf{z}_k\right)}{\partial \mathbf{z}^i} > 0, i \notin \mathcal{I}_k^{\mathrm{Proj}} \right\}, \quad \mathcal{I}_{2,k} = \left\{ i \mid \frac{\partial \ell\left(\mathbf{z}_k\right)}{\partial \mathbf{z}^i} \leq 0, i \notin \mathcal{I}_k^{\mathrm{Proj}} \right\}.$$

For all $i \in \mathcal{I}_{1,k}$, we must have $\mathbf{z}_k^i > \varepsilon_k$, otherwise we would have $i \in \mathcal{I}_k^{\mathrm{Proj}}$. Since $\left| \bar{\mathbf{z}} - [\bar{\mathbf{z}} - M\nabla\ell(\bar{\mathbf{z}})]^{\mathrm{Proj}} \right| \neq 0$, we must have $\lim_{k \to 0, k \in K} \inf \varepsilon_k > 0$ and $\varepsilon_k > 0$ for all $k$. Let $\bar{\varepsilon} > 0$ be such that $\bar{\varepsilon} \leq \varepsilon_k$ for all $k \in K$, and let $B$ denote a constant such that $\left| p_k^i \right| \leq B$ for all $i$ and $k \in K$. Then for all $\alpha \in [0, \bar{\varepsilon}/B]$, we have $\mathbf{z}_k^i(\alpha) = \mathbf{z}_k^i - \alpha p_k^i$ for all $i \in I_{i,k}$, so it follows that:

$$\sum_{i \in I_{1,k}} \frac{\partial \ell\left(\mathbf{z}_k\right)}{\partial \mathbf{z}^i} \left[ \mathbf{z}_k^i - \mathbf{z}_k^i(\alpha) \right] = \alpha \sum_{i \in I_{1,k}} \frac{\partial \ell\left(\mathbf{z}_k\right)}{\partial \mathbf{z}^i} p_k^i \quad \forall \alpha \in \left[ 0, \frac{\bar{\varepsilon}}{B} \right].$$

Also, for all $\alpha \geq 0$, we have $\mathbf{z}_k^i - \mathbf{z}_k^i(\alpha) \leq \alpha p_k^i$, and since $\frac{\partial \ell(\mathbf{z}_k)}{\partial \mathbf{z}^i} \leq 0$ for all $i \in I_{2,k}$, we obtain

$$\sum_{i \in I_{2,k}} \frac{\partial \ell\left(\mathbf{z}_k\right)}{\partial \mathbf{z}^i} \left[ \mathbf{z}_k^i - \mathbf{z}_k^i(\alpha) \right] \geq \alpha \sum_{i \in I_{2,k}} \frac{\partial \ell\left(\mathbf{z}_k\right)}{\partial \mathbf{z}^i} p_k^i.$$

Combining these two equations above, we obtain

$$\sum_{i \in \mathcal{I}_k} \frac{\partial \ell\left(\mathbf{z}_k\right)}{\partial \mathbf{z}^i} \left[ \mathbf{z}_k^i - \mathbf{z}_k^i(\alpha) \right] \geq \alpha \sum_{i \in \mathcal{I}_k} \frac{\partial \ell\left(\mathbf{z}_k\right)}{\partial \mathbf{z}^i} p_k^i \quad \forall \alpha \in \left[ 0, \frac{\bar{\varepsilon}}{B} \right].$$

For all $\alpha \geq 0$, we also have

$$\left| \mathbf{z}_k^i - \mathbf{z}_k^i(\alpha) \right| \leq \alpha \left| p_k^i \right| \quad \forall i = 1, \cdots, n.$$

Furthermore, it is easily seen using Eq. (13) that there exists $\lambda > 0$ such that

$$\sum_{i \in \mathcal{I}_k} \left( p_k^i \right)^2 \leq \lambda \sum_{i \in \mathcal{I}_k} \frac{\partial \ell\left(\mathbf{z}_k\right)}{\partial \mathbf{z}^i} p_k^i \quad \forall k \in K.$$

Combining the last two relations, we obtain for all $\alpha \geq 0$, it holds that:

$$\sum_{i \in \mathcal{I}_k} \left| \mathbf{z}_k^i - \mathbf{z}_k^i(\alpha) \right|^2 \leq \alpha^2 \lambda \sum_{i \in \mathcal{I}_k} \frac{\partial \ell\left(\mathbf{z}_k\right)}{\partial \mathbf{z}^i} p_k^i \quad \forall k \in K.$$

We now obtain for all $\alpha \in [0, (\bar{\varepsilon}/B)]$ and $k \in K$, it holds that:

$$\ell\left(\mathbf{z}_k\right) - \ell\left[\mathbf{z}_k(\alpha)\right] \geq \left( \alpha - \frac{\alpha^2 \lambda L}{2} \right) \sum_{i \in \mathcal{I}_k} \frac{\partial \ell\left(\mathbf{z}_k\right)}{\partial \mathbf{z}^i} p_k^i + \left( 1 - \frac{\alpha \bar{\lambda}_2 L}{2} \right) \sum_{i \in \mathcal{I}_k} \frac{\partial \ell\left(\mathbf{z}_k\right)}{\partial \mathbf{z}^i} \left[ \mathbf{z}_k^i - \mathbf{z}_k^i(\alpha) \right].$$

Suppose $\alpha$ is chosen such that:

$$0 \leq \alpha \leq \frac{\bar{\varepsilon}}{B}, \quad 1 - \frac{\alpha \lambda L}{2} \geq \sigma, \quad 1 - \frac{\alpha \overline{\lambda_2} L}{2} \geq \sigma, \quad \alpha \leq 1;$$

or equivalently,

$$0 \leq \alpha \leq \min \left\{ \frac{\bar{\varepsilon}}{B}, \frac{2(1-\sigma)}{\lambda L}, \frac{2(1-\sigma)}{\bar{\lambda}_2 L}, 1 \right\}. \tag{23}$$

Then for all $k \in K$, we have:

$$\ell\left(\mathbf{z}_k\right) - \ell\left[\mathbf{z}_k(\alpha)\right] \geq \sigma \left\{ \alpha \sum_{i \in \mathcal{I}_k} \frac{\partial \ell\left(\mathbf{z}_k\right)}{\partial \mathbf{z}^i} p_k^i + \sum_{i \in \mathcal{I}_k} \frac{\partial \ell\left(\mathbf{z}_k\right)}{\partial \mathbf{z}^i} \left[\mathbf{z}_k^i - \mathbf{z}_k^i(\alpha)\right] \right\}.$$

This means that if Eq. (23) is satisfied with $\beta^m = \alpha$, then the inequality Eq. (11) of the Armijo-like rule will be satisfied. It follows from the way the stepsize is reduced that $\alpha_k$ satisfies:

$$\alpha_k \geq \beta \min \left\{ \frac{\bar{\varepsilon}}{B}, \frac{2(1-\sigma)}{\lambda L}, \frac{2(1-\sigma)}{\bar{\lambda}_2 L}, 1 \right\} \quad \forall k \in K.$$

This contradicts Eq. (19). From Eq. (14), since $f$ is twice differentiable on $\{\mathbf{z} \mid |\mathbf{z} - \mathbf{z}^*| < \delta\}$, it follows that there exist scalars $L > 0$ and $\delta_1 \in (0, \delta]$ such that for all $\mathbf{z}, \mathbf{y}$ with $|\mathbf{z} - \mathbf{z}^*| \leq \delta$ and $|\mathbf{y} - \mathbf{z}^*| \leq \delta_1$, we have

$$|\nabla \ell(\mathbf{z}) - \nabla \ell(\mathbf{y})| \leq L|\mathbf{z} - \mathbf{y}|.$$

Also, for $\mathbf{z}_k$ sufficiently close to $\mathbf{z}^*$, the scalar $w_k = \left| \mathbf{z}_k - \left[\mathbf{z}_k - M\nabla \ell\left(\mathbf{z}_k\right)\right]^{\mathrm{Proj}} \right|$ is arbitrarily close to zero. Thus we have:

$$\left[ \mathbf{z}_k^i - \mu^i \frac{\partial \ell\left(\mathbf{z}_k\right)}{\partial \mathbf{z}^i} \right]^{\mathrm{Proj}} = 0 \quad \forall i \in B\left(\mathbf{z}^*\right),$$

where $\mu^i$ is the $i$-th diagonal element of $M$. It follows that for $\mathbf{z}_k$ sufficiently close to $\mathbf{z}^*$, we have

$$\mathbf{z}_k^i \leq w_k = \varepsilon_k \quad \forall i \in B\left(\mathbf{z}^*\right), \text{ while } \mathbf{z}_k^i > \varepsilon_k \quad \forall i \notin B\left(\mathbf{z}^*\right).$$

This implies that there exists $\delta_2 \in (0, \delta_1]$ such that

$$B\left(\mathbf{z}^*\right) = \mathcal{I}_k^{\mathrm{Proj}} \quad \forall k \text{ such that } |\mathbf{z}_k - \mathbf{z}^*| \leq \delta_2.$$

Also, there exist scalars $\bar{\varepsilon} > 0$ and $\delta_3 \in (0, \delta_2]$ such that

$$\mathbf{z}_k^i > \bar{\varepsilon} \quad \forall i \notin B\left(\mathbf{z}^*\right) \text{ and } k \text{ such that } |\mathbf{z}_k - \mathbf{z}^*| \leq \delta_3.$$

By essentially repeating the argument in the proof of above that, we find that there exists a scalar $\bar{\alpha} > 0$ such that

$$\alpha_k \geq \bar{\alpha} \quad \forall k \text{ such that } |\mathbf{z}_k - \mathbf{z}^*| \leq \delta_3.$$

It follows that

$$0 < \bar{\lambda}_1 \frac{\partial \ell\left(\mathbf{z}_k\right)}{\partial \mathbf{z}^i} \leq p_k^i \quad \forall i \in B\left(\mathbf{z}^*\right) \text{ and } k \text{ such that } |\mathbf{z}_k - \mathbf{z}^*| \leq \delta_3,$$

While there exists a scalar $\lambda > 0$ such that

$$\sum_{i \in B(\mathbf{z}^*)} \left|p_k^i\right|^2 \leq \lambda \sum_{i \in B(\mathbf{z}^*)} \left| \frac{\partial \ell\left(\mathbf{z}_k\right)}{\partial \mathbf{z}^i} \right|^2 \quad \forall k \text{ such that } |\mathbf{z}_k - \mathbf{z}^*| \leq \delta_3.$$

Since $\frac{\partial \ell(\mathbf{z}^*)}{\partial \mathbf{z}^i} > 0$ for $\forall i \in B\left(\mathbf{z}^*\right)$ and $\frac{\partial f(\mathbf{z}^*)}{\partial \mathbf{z}^i} = 0$ for $\forall i \notin B\left(\mathbf{z}^*\right)$, it follows that there exists a scalar $\delta_4 \in (0, \delta_3]$ such that

$$B\left(\mathbf{z}^*\right) = B\left(\mathbf{z}_{k+1}\right) \text{ and } |\mathbf{z}_{k+1} - \mathbf{z}^*| \leq \delta_3; \quad \forall k \text{ such that } |\mathbf{z}_k - \mathbf{z}^*| \leq \delta_4.$$

Thus we have

$$B\left(\mathbf{z}^*\right) = B\left(\mathbf{z}_{k+1}\right) = I_{k+1}^{\mathrm{Proj}} \quad \forall k \text{ such that } |\mathbf{z}_k - \mathbf{z}^*| \leq \delta_4.$$

This to say, when $|\mathbf{z}_k - \mathbf{z}^*| \leq \delta_4$, we have $|\mathbf{z}_{k+1} - \mathbf{z}^*| \leq \delta_3$, $B\left(\mathbf{z}^*\right) = B\left(\mathbf{z}_{k+1}\right)$, and the $(k+1)$-st iteration of the algorithm reduces to an iteration of an unconstrained minimization algorithm on the subspace of binding constraints at $\mathbf{z}^*$. Given the well-known quadratic convergence rate of the unconstrained Newton method, we thus obtain a guarantee on the convergence rate.

# D. Experiments

The experiments are run on a Nvidia Tesla A100 GPU and two Intel Xeon Platinum 8358 CPUs. Besides, the RL results (Fig. 4) are obtained by using the stable-baselines3 library (Raffin et al., 2021).

## D.1. Synthetic Data

In this section, we provide details about the Market-Manage data. The variables included in the generation process are:

- $\text{Feature}_{\text{our}}$: The feature used to predict the raw cost of our market;

- $\text{Feature}_{\text{cpt}}$: The feature used to predict the raw cost of the competitor market;

- $\text{C}_{\text{our}}$: The raw cost of our market;

- $\text{C}_{\text{cpt}}$: The raw cost of the competitor market;

- $\text{P}_{\text{our}}$: The product price of our market;

- $\text{P}_{\text{cpt}}$: The product price of the competitor market;

- NCT: Customer numbers of our market;

- TPF: Total profit of our market.

The rehearsal graph for the variables is illustrated in Fig. 7. The presumed actionable variables that can be altered by the manager are $\text{C}_{\text{our}}$ and $\text{P}_{\text{our}}$.

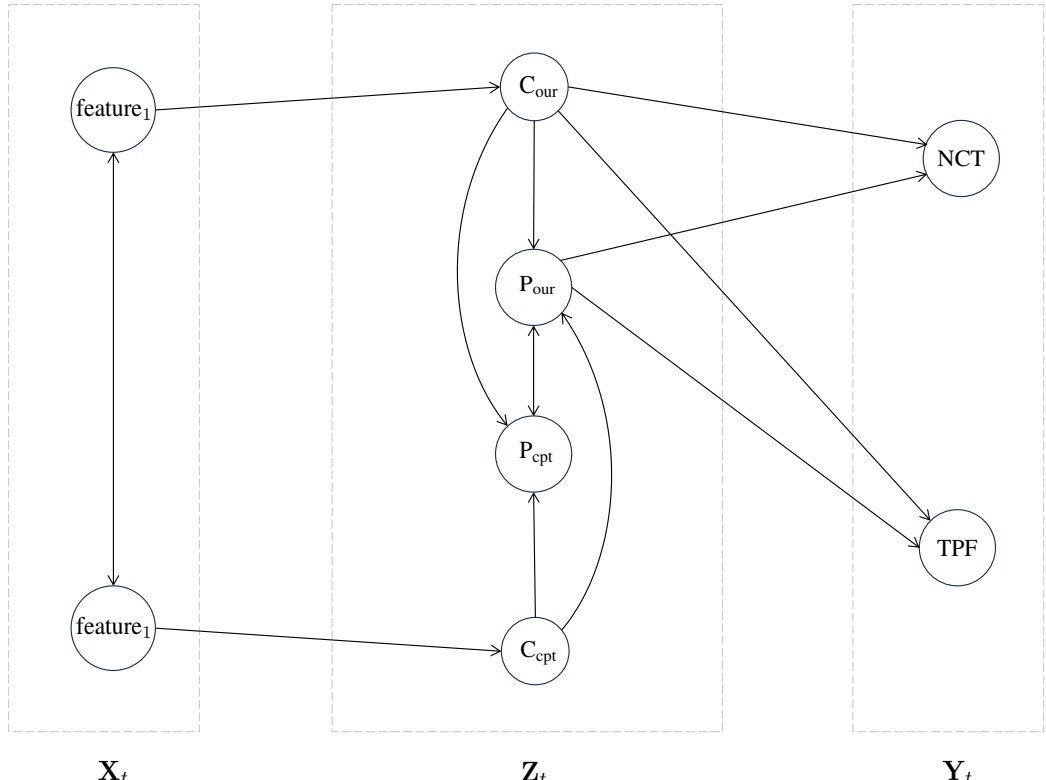

*Figure 7.* The rehearsal graph for market-manage data.

## D.2. Bermuda Data

In this section, we provide details about the Bermuda data. The Bermuda data is an environment dataset that involves some environmental variables in Bermuda (Courtney et al., 2017). The variables included in the generation process are:

- Light: Light levels at the bottom;

- Temp: Temperature at the bottom;

- Sal: Sea surface salinity;

- DIC: Dissolved inorganic carbon of seawater;

- TA: Total alkalinity of seawater;

- $\Omega_A$: Saturation with respect to aragonite in seawater;

- Chla: Chlorophyll-a at sea surface;

- Nut: PC1 of $NH_4$, $NiO_2 + NiO_3$, $SiO_4$;

- pHsw: pH of seawater;

- $CO_2$: $P_{CO_2}$ of seawater;

- NEC: Net ecosystem calcification.

The rehearsal graph for the variables is illustrated in Fig. 8. The presumed actionable variables that can be altered by the decision-maker are DIC, TA, $\Omega_A$, Chla, and Nut according to Aglietti et al. (2020); Qin et al. (2023); Du et al. (2024).

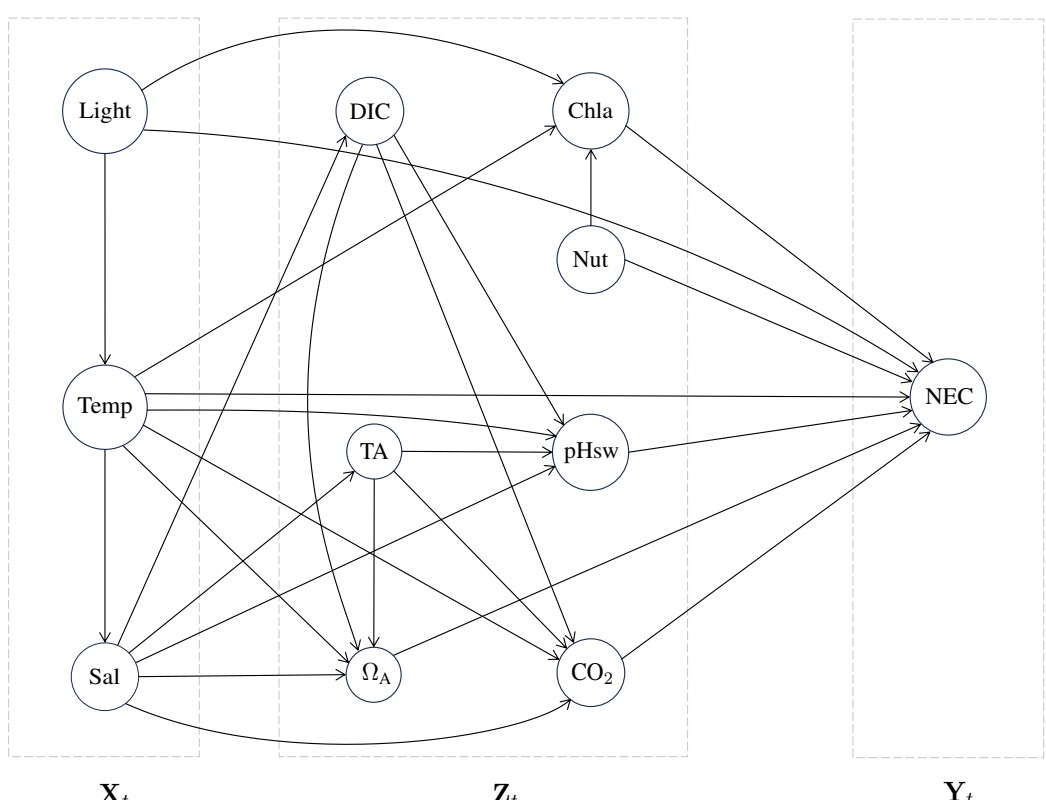

*Figure 8.* The rehearsal graph for Bermuda data.

