# OpenReview forum: "Enabling Optimal Decisions in Rehearsal Learning under CARE Condition"
_ICML.cc/2025/Conference — ICML 2025 poster_

### Official Review · Reviewer_UMM3 · 2025-02-24

**Overall Recommendation:** 3

**Summary:**

The paper introduces a CAnonical REctangle (CARE) condition for the Avoiding Undesired Future (AUF) problem. Under this CARE condition, along with additional assumptions on the problem structure and the noise term, the AUF problem can be reformulated as a convex optimization problem. The authors propose a projection-Newton-based algorithm with a provable sublinear convergence rate and extend the approach to cases where the CARE condition does not hold. Furthermore, for the special case where the target variable $Y$ is single-dimensional, the paper derives a closed-form solution that significantly reduces time complexity. Numerical experiments on both synthetic and real-world datasets are provided to demonstrate the effectiveness and efficiency of the proposed algorithms.

**Claims And Evidence:**

Yes.

**Essential References Not Discussed:**

Not sure.

**Experimental Designs Or Analyses:**

While the overall experimental design seems sound, the paper does not provide sufficient details on the experimental setup in its appendix. For example, the procedure for generating the synthetic data is not clearly explained.

**Methods And Evaluation Criteria:**

Yes.

**Other Comments Or Suggestions:**

None.

**Other Strengths And Weaknesses:**

Strengths: The paper is well written and provides clear remarks and visualizations that effectively illustrate the theoretical results.

Weaknesses: The discussion regarding the additive noise is limited; it appears that the results may only hold under Gaussian noise assumptions. Additionally, the experimental section would be clearer with more detailed descriptions of the experimental setup.

**Questions For Authors:**

a) If the additive noise in equation (1) does not follow a Gaussian distribution, do the proposed methods remain (numerically) effective? A discussion on potential performance variations or issues under non-Gaussian noise, or a potential extension method which can be applied in non-Gaussian noises would help clarify the broader applicability of the results.

b) Related to the previous question, if Theorem 3.6 is valid only under the assumption of Gaussian noise, are there other families of distributions (e.g., those within the exponential family) for which the results could still hold? The limitation of Gaussian noises is my major concern.

c) Could you provide more detailed information on how the synthetic data was generated, as well as a summary of the characteristics of the real-world dataset used in your experiments?

**Relation To Broader Scientific Literature:**

The results presented can be applied to general AUF problems under specific conditions.

**Theoretical Claims:**

I reviewed the proofs of Theorem 3.6 and Theorem 3.8, and they appear to be correct to the best of my knowledge.

---

> ### Author Rebuttal · Authors · 2025-03-31
>
> Thanks for the valuable feedback! We hope our responses can address your concerns.
>
> **Q1.** Extension for non-Gaussian noise.
>
> **A1.** Thanks for your insightful question. We would like to clarify that the Gaussian noise assumption is primarily used to establish theoretical guarantees. For cases where this assumption does not hold, AUF can also be addressed effectively by:
>
> - **Gaussian approximation.** It is common to approximate irregular distributions using a Gaussian distribution. E.g., Laplace's approximation [1] can approximate unimodal distributions by fitting a Gaussian distribution centered at the mode. To validate the empirical effectiveness of our method under non-Gaussian noise, we provide additional experiments as shown in Fig. 1 of [Anon. Link](https://default-anno-bucket.s3.us-west-1.amazonaws.com/rebuttal.pdf).
>
> - **A numerical solution.** An alternative empirical approach can be used to solve AUF in Eq. (2) by training a sampler (e.g., normalizing flow) for the potentially non-Gaussian noise using residuals. This method involves sampling $n$ noise realizations from the trained sampler and selecting $z^\xi$ to maximize the number of instances where $Y\in S$. This can be formulated as the following mixed-integer linear program:
> $$\begin{aligned}\max_{e_i\in\{0,1\}, z^\xi}\ &\sum_{i=1}^n e_i\\\\s.t.\ &M(Ax+Bz^\xi+C\epsilon_i)-d\leq(1-e_i)\alpha,\ i\in[n]\end{aligned}$$
> where $e_i$ indicates whether $i$-th sample is successful, $\alpha$ is a sufficiently large constant vector to tolerate failed samples, $M,d,A,B,C$ are defined in Thm. 3.6. This approach can be empirically applied with non-Gaussian noise.
>
> We will incorporate this discussion into the revised paper. Thanks!
>
> ---
>
> **Q2.** Potential distribution family making Thm. 3.6 valid.
>
> **A2.** Thanks for your question. The validity of Thm. 3.6 stems from the fact that Gaussian belongs to both elliptical and exponential families. Hence, the decomposition (Eq. 8) and the log-concavity (lines 611–628) can be established.
>
> The main challenge in extending these results to general distributions, such as the exponential family, is that simultaneously satisfying both the decomposition and log-concavity is difficult. Additionally, the PDF of a general distribution can be complex, limiting the applicability of existing analytical techniques. Recognizing the importance of extending theoretical results to non-Gaussian cases, we are currently investigating a similar theoretical foundation for the log-concave family (including Gaussian, uniform, and Laplace distributions, etc.) as part of our future work. In this case, we find that techniques used in this paper are insufficient, and the Prékopa–Leindler theorem [2] may provide a useful tool for proving log-concavity of generalized distribution families. However, explicit characterization for these distributions remains a significant challenge.
>
> Finally, we would like to emphasize that proving theoretical optimality for probabilistic optimization is challenging, even under Gaussian noise. Establishing theoretical guarantees under Gaussian noise provides insights that extend beyond Gaussian scenarios and is a common practice in studies on previous rehearsal learning [3, 4], time series analysis [5], and control problems [6], among others.
>
> We will incorporate this discussion in the revised paper. Thanks!
>
> ---
>
> **Q3.** Detailed data information.
>
> **A3.** Thanks for your question.
> - **For synthetic data**. Let $V=[feature_1,feature_2,C_{our},C_{cpt},P_{our},P_{cpt},NCT,TPF]$ with variables defined in Fig. 5. The data generation follows $V=PV+\epsilon,\epsilon\sim\mathcal{N}(0,\Sigma)$:
> $$P=\begin{pmatrix}0&0&0&0&0&0&0&0\\\\0&0&0&0&0&0&0&0\\\\10&0&0&0&0&0&0&0\\\\0&10&0&0&0&0&0&0\\\\0&0&2.0&0.4&0&0&0&0\\\\0&0&0.5&1.3&0&0&0&0\\\\0&0&1.6&0&-0.9&0&0&0\\\\0&0&-1&0&0.9&0&0&0\end{pmatrix},\Sigma=10^{-2}\begin{pmatrix}4&0&0&0&0&0&0&0\\\\0&4&0&0&0&0&0&0\\\\0&0&6&0&0&0&0&0\\\\0&0&0&3&16&0&0&0\\\\0&0&0&16&6&0&0&0\\\\0&0&0&0&0&6&0&0\\\\0&0&0&0&0&0&4&0\\\\0&0&0&0&0&0&0&12\end{pmatrix}.$$
> As $feature_{1,2}$ have no parents, their covariance can be marginalized thus omitted. Parameters are identical to those used in the code provided in Supp. Material and experimental results can be reproduced by running the code.
>
> - **For real-world data**. The dataset records values of environment variables in Bermuda, and the decision target is to maintain a high NEC (net coral ecosystem calcification). Due to space limits, we welcome further questions on the dataset and will provide additional details in Appx. E.2 of the revised paper.
>
> Thanks again!
>
> ---
>
> **References:**
>
> [1] Pattern Recognition and Machine Learning, 2006
>
> [2] On logarithmic concave measures and functions, 1973
>
> [3] Avoiding Undesired Future with Minimal Cost in Non-Stationary Environments, NeurIPS 2024
>
> [4] Rehearsal Learning for Avoiding Undesired Future, NeurIPS 2023
>
> [5] Time series analysis and its applications, 2000
>
> [6] Online Linear Quadratic Control, ICML 2018

---

### Official Review · Reviewer_XTv5 · 2025-03-16

**Overall Recommendation:** 4

**Summary:**

This paper addresses the AUF (Avoiding Undesired Future) problem in machine learning decision-making, where the goal is to identify actions that prevent undesirable outcomes predicted by ML models.  It introduces the CARE condition (CAnonical REctangle), a novel assumption under which the AUF probability—i.e., the probability that a post-decision outcome falls within a desired region—can be explicitly expressed and transformed (via a negative log operation) into a convex function. This convexity enables efficient optimization. They present a projection-Newton algorithm that achieves superlinear convergence to the optimal decision alteration and an inner embedding technique for cases where the CARE condition does not hold. Additionally, the paper provides a closed-form solution when the outcome is a singleton, significantly reducing time complexity. The experimental results on both synthetic and real-world datasets demonstrate that the proposed approach not only improves the AUF probability but also enhances computational efficiency compared to existing rehearsal learning and reinforcement learning methods.

**Claims And Evidence:**

Most of the claims are well supported by both theoretical derivations and empirical results. However, the assertion that the CARE condition is prevalent in real-world scenarios deserves additional discussion. The authors claim that this condition particularly holds when the dimensions of Y, such as Y₁ and Y₂, are mutually independent. In many practical situations, however, these objectives tend to be dependent—improvements in one may lead to declines in another. For example, in an autonomous drone delivery system, reducing the risk of package loss may require higher altitudes or longer routes, which in turn increase delivery times; similarly, in healthcare treatment planning, enhancing treatment efficacy often comes with more severe side effects. In portfolio optimization, striving to maximize returns usually entails accepting higher risk, illustrating that objectives are frequently interdependent rather than mutually independent.

This dependency calls for further clarification or evidence to convincingly support the claim regarding the prevalence of the CARE condition, which underpins the paper.

**Generalization Beyond CARE via Inner CARE Embedding:**
While the paper introduces an inner CARE embedding technique to handle scenarios where the CARE condition does not naturally hold, the evidence here is somewhat less extensive. Although Propositions 3.10 and 3.11 provide a theoretical basis and a demonstration for circular regions, the empirical evaluation of this generalization is more limited. In practical settings where the desired region S is irregular or non-canonical, additional experiments or case studies might be necessary to fully convince readers of its broad applicability.

**Essential References Not Discussed:**

Not that I know of.

**Experimental Designs Or Analyses:**

The experimental design and analyses looks good to me.

**Methods And Evaluation Criteria:**

The proposed methods are well-aligned with the AUF problem, featuring a projection-Newton algorithm, an inner CARE embedding for irregular cases, and a closed-form solution for unidimensional outcomes. The evaluation criteria—including AUF probability, success frequency over multiple rounds, and decision-making time—directly reflect the challenges of immediate, interaction-free decisions in rehearsal learning. Additionally, the use of both synthetic and real-world datasets, along with comparisons to state-of-the-art methods, provides a comprehensive and practical benchmark for the proposed approach.

**Other Comments Or Suggestions:**

N/A

**Other Strengths And Weaknesses:**

The paper is solid, and clear, with good theoretical and experimental results. However, the authors should backup the CARE condition with more real-world examples.

**Questions For Authors:**

N/A

**Relation To Broader Scientific Literature:**

Regarding the rehearsal learning and AUF problem characterized by linear structural equations scientific literature, this paper introduces the CARE condition which imposes the concave structure on the logged AUF probability, making this problem more tractable and paving the way to structured algorithmic solution like the projection-Newton method proposed herein.

**Theoretical Claims:**

The theoretical proofs looks sound to me.

---

> ### Author Rebuttal · Authors · 2025-03-31
>
> Thanks for your valuable feedback and appreciation of our work! We hope that our responses can address your concerns.
>
> **W1.** Further Discussion on the CARE Condition.
>
> **A1.** Thanks for your insightful question. In practice, the dimensions of $\mathbf{Y}$ are often dependent, meaning that the common region defined as {$a_i \leq Y_i \leq b_i$}, $i=1,\dots,|\mathbf{Y}|$ does not necessarily form a canonical rectangle w.r.t. the covariance matrix of $\mathbf{Y}$. However, the CARE condition remains useful because the desired region $\mathcal{S}$ can be manually specified by the decision-maker, as decisions are made after obtaining some system information. The overall decision-making process in rehearsal learning proceeds as follows:
>
> - **Step 1.** Initially, historical observational data samples are available, enabling estimation of the underlying system parameters $\theta$.
> - **Step 2.** Consequently, matrices such as $C$ and $\Sigma$ (for Def. 3.5 of the CARE condition) are available before decision-making. Since the desired properties of the target $\mathbf{Y}$ are typically determined by the decision-maker, the target region can be manually adjusted based on the estimated $C$ and $\Sigma$ to ensure compliance with the CARE condition.
> - **Example.** In portfolio optimization, defining a desired region such as $\{a_1 \leq \text{returns} \leq b_1\}$ and $\{a_2 \leq \text{risk} \leq b_2\}$ ensures a balanced strategy with both favorable returns and controlled risk. Alternatively, using desired region such as $\{a \leq \alpha\cdot\text{returns}+\beta\cdot\text{risk}\leq b\}$ and $\{c \leq \beta\cdot\text{returns}-\alpha\cdot\text{risk} \leq d\}$ (where $\alpha, \beta\neq 0$ and depend on $C$, $\Sigma$) can also achieve a balanced strategy while satisfying the CARE condition, allowing for a theoretically optimal solution.
>
> Hence, the CARE condition can be satisfied by appropriately adjusting the desired region if such adjustments are feasible. Additionally, the inner CARE embedding of {$a_i \leq Y_i \leq b_i$} can be computed using techniques for identifying axis-parallel rectangles within polygons [1]. By defining a basis aligned with the specific matrix space, the CARE embedding of the polygon {$a_i \leq Y_i \leq b_i$} can be derived. Additional experimental results provide support of this argument as detailed in **A2**.
>
> We will incorporate this discussion in the revised version. Thanks!
>
> ---
>
> **W2.** Additional case studies of inner CARE embedding.
>
> **A2.** Thanks for your thoughtful question. We conducted additional experiments for cases where the dimensions of $\mathbf{Y}$ remain **dependent** even after alterations, demonstrating that our proposed inner CARE embedding method is effective and flexible. Specifically, the experiments include two types of desired regions:  (i) Circular desired region, with the inner CARE embedding computed using Eq. (4); (ii) Axis-aligned rectangular region {$a_i \leq Y_i \leq b_i$}, with the inner CARE embedding computed using [1]. The visualizations are presented in Fig. 2 of [Anon. Link](https://default-anno-bucket.s3.us-west-1.amazonaws.com/rebuttal.pdf), and the detailed performance, measured by AUF probability, is listed below:
>
> | Region types|      No action      | QWZ23 [2]        | MICNS [3]        | Ours             |
> | --------------------------- | :--------------: | ---------------- | ---------------- | ---------------- |
> | Circular region| $0.010\pm 0.036$ | $0.961\pm 0.013$ | $0.957\pm 0.024$ | $0.983\pm 0.018$ |
> | {$a_i \leq Y_i \leq b_i$} | $0.004\pm 0.021$ | $0.885\pm 0.028$ | $0.893\pm 0.044$ | $0.922\pm 0.083$ |
>
> These results show that our proposed generalization method, i.e., inner CARE embedding, is effective in several irregular or non-canonical scenarios.
>
> Additionally, we would like to emphasize that finding a maximal axis-parallel inner rectangle for an irregular region is a challenging geometric problem [1], especially in high-dimensional spaces. Hence in practice, one can instead identify an inner canonical rectangle (not necessarily the maximal one) and optimize using this inner rectangle as a surrogate, which still enables a deterministic transformation in Cor. 3.7. This approach remains valid, as any inner region of the original retains a probability mass less than or equal to that of the original region, ensuring consistency with Prop. 3.10 due to the non-negativity of the probability density function.
>
> We will incorporate these results and discussions into the revised paper. Thanks!
>
> ---
>
> **References:**
>
> [1] Finding the largest area axis-parallel rectangle in a polygon, Computational Geometry 1997.
>
> [2] Rehearsal Learning for Avoiding Undesired Future, NeurIPS 2023.
>
> [3] Avoiding Undesired Future with Minimal Cost in Non-Stationary Environments, NeurIPS 2024.
>
> ---
>
> We also take this opportunity to sincerely thank you for the careful review. Your suggestions are important for further improving the paper. Thanks again!

---

### Official Review · Reviewer_YMZh · 2025-03-16

**Overall Recommendation:** 3

**Summary:**

The paper proposes an algorithm for decision making that helps avoid undesirable future (AUF), i.e., increasing the AUF probability. The new algorithm is shown to reduce time complexity compared to prior work and has showed performance improvement compared to a few baselines.

**Claims And Evidence:**

The theoretical claims in the paper seem to be supported by proper arguments. There are also limited empirical evidence supporting the proposed method.

**Essential References Not Discussed:**

NA

**Experimental Designs Or Analyses:**

The experimental design seems reasonable though I think it might be good to add a few more baseline methods.

**Methods And Evaluation Criteria:**

The main empirical evaluation criterion is the probability of AUF, as measured by the empirical results.

**Other Comments Or Suggestions:**

NA

**Other Strengths And Weaknesses:**

The paper proposes a more efficient algorithm compared to prior work, which is very desirable and this has been established theoretically. In practice there are also empirical improvements compared to prior work. I think the main weakness is how the methodology here relates more broadly to the RL community, both in terms of the methodology relevance and application relevance.

**Questions For Authors:**

=== *relevance to RL literature* ====

From the looks of the definition of AUF, it seems that we can think of it as a RL problem where success gets $r=1$ and failure gets $r=0$, by maximizing the average reward we essentially maximize AUF. I wonder if framing the problem this way can help bridge the work's relevance to more general RL community. Do the authors agree with this characterization?

=== *RL algorithm baseline* ===

If the above characterization is proper, then I wonder whether there can be more RL related baselines to compare against in Fig 4 (c), such as reinforce policy gradient.

I also find that the methodology proposed in this work has a rather strong "model based|" flavor, in that in order to obtain the solution we need to already know the graphical model behind the transition dynamics. This access to the ground truth model provides an advantage to the proposed method and I wonder what happens if the assumed model deviates from the real world domain, can we characterize the performance regret in that case? and what if we get to update the model based on real life applications? these ablations or discussions or bring the proposed method closer to the problems that RL community is well-versed in.

=== *time complexity* ===

As a minor point, I wonder what $p$ stands for in the time complexity comparison in Fig 1. I also assume that this complexity is obtained under assuming access to the ground truth graphical model of the problem, and I wonder what is the sensitivity of the theoretical result to the correctness of the graphical model (ie what happens if the model is $\epsilon$ away from the ground truth model. I think characterizing those will be more meaningful for practitioners.

**Relation To Broader Scientific Literature:**

I think the work is related to reinforcement learning and especially its deployment in real applications.

**Theoretical Claims:**

I did not check the correctness of the proof.

---

> ### Author Rebuttal · Authors · 2025-03-31
>
> Thanks for your detailed feedback! We hope our responses address your concerns.
>
> **Q1.** Relevance to RL research.
>
> **A1.** Thanks for your insightful question. Below, we clarify the connection between RL and AUF problem and explain distinctions.
>
> - **Connection between RL and AUF.** When interactions are available, AUF can be formulated as an MDP (or reduced to a Bandit if no state transitions) and solved using RL methods. Specifically, at round $t$, state $x_t$ is observed, then an action is taken by altering $z_t$, and the environment provides a reward: $r=1$ if $y_t\in S$ else $r=0$. This aligns with our RL baselines (Tab. 2&3) and additional experiments in **A2**.
>
> - **Distinctions between RL and rehearsal learning.** Rehearsal learning focuses on a specialized decision-making setting where interactions are limited or even unavailable. In this case, variables **X,Z,Y** follow a structured generative process parameterized by $\theta$ (in MDP formulation, transition dynamics and reward function also depend on $\theta$). Leveraging this fine-grained structure, reliable decisions can be made using only a small set of observational samples (for estimating $\hat\theta$) without necessarily requiring interactions with the environment. In contrast, online RL methods rely on extensive interactions for effective policy learning. While offline/hybrid online-offline RL might seem applicable, direct applications would also be unsuitable, as actions significantly shift the distribution of **Y** as discussed in Sec. 2, rendering the reward functions different between offline and online data.
>
> While RL/MDP provides a general framework for decision-making, real-world interactions can be costly or even unethical. E.g., in healthcare, doctors cannot freely experiment with untested treatments. In such cases, leveraging structural knowledge enables effective policy without interaction-based exploration while also enhancing interpretability.
>
> Finally, integrating a rehearsal-learning policy as an initialization for online RL could serve as a potential bridge between our work and the broader RL community, offering a promising yet challenging direction to improve RL’s sample efficiency in certain cases.
>
> We will incorporate this discussion into the revised paper. Thanks!
>
> ---
>
> **Q2.** RL baseline&Discussion on $\hat{\theta}$.
>
> **A2.** Thanks for your question. We conduct additional experiments in Fig. 3 of [Anon. Link](https://default-anno-bucket.s3.us-west-1.amazonaws.com/rebuttal.pdf) to show that policy gradient  RL methods can be effective for AUF when sufficient interactions are available. Note that Fig. 4c in paper illustrates the performance of rehearsal methods w.r.t. the number of ***offline observational samples***, making it unsuitable for evaluating RL methods (including offline/hybrid online-offline RL as discussed in **A1**).
>
> Furthermore, we address implications of using an $\epsilon$-approximate model $\hat{\theta}$, i.e., $||\hat{\theta}-\theta||\leq \epsilon$:
>
> - **Empirical validation.** As shown in [1], $||\hat{\theta}-\theta||$ decreases as the number of observational samples increases. Hence, Fig. 4c shows that our approach's performance improves as $\epsilon$ decreases in practice.
>
> - **Theoretical analysis.** We would like to clarify that our theoretical guarantees are established on the AUF probability conditioned on $\hat\theta$ rather than on the true parameter $\theta$, as discussed below Thm. 3.8. Moreover, deriving a regret bound similar to those in RL literature is generally challenging because the rehearsal-learning policy is obtained by solving a probabilistic optimization (Eq. 2). In this case, although the output action $z$ depends on $\theta$, properties such as L-Lipschitz continuity are extremely difficult to establish, making regret analysis nontrivial. However, if the function $\ell \circ z(\cdot)$ can be assumed to be L-Lipschitz continuous, then the following bound can be derived:
> $$\ell(z(\hat{\theta}))-\ell(z^*)\leq L||\hat{\theta}-\theta||_2\lesssim\mathcal{O}(\frac{1}{\sqrt{n}}),$$
> where $n$ is the number of observational samples. The last inequality follows from [1]. Finally, online updating of $\hat{\theta}$ can also be incorporated in our approach via an offline parameter-update step after each decision round.
>
> We will refine this discussion in the revised version. Thanks!
>
> ---
>
> **Q3.** The meaning of $p$.
>
> **A3.** Thanks for your feedback. In Tab. 1,  $p$ represents the dimensionality of actionable variables $z^\xi$.
>
> Rehearsal learning has two-stages: (i) estimate $\theta$ from historical samples; (ii) make decisions after observing $x$. Only stage (ii) requires immediate actions, and Tab. 1 reports its time complexity. The sensitivity of theoretical results is discussed in **A2**.
>
> We will incorporate this clarification in the revised version. Thanks again!
>
> ---
>
> **References:**
>
> [1] Avoiding Undesired Future with Minimal Cost in Non-Stationary Environments, NeurIPS 2024.

---

### Decision · Program_Chairs · 2025-05-01

**Decision:**

Accept (poster)

**Comment:**

The paper received mildly positive reviews. There are strengths and weaknesses:

**Strengths**

- Novelty: Introduces the CARE condition to make the AUF problem convex and tractable.

- Clarity: Well-written with clear visualizations.

- Practical Performance: Outperforms baselines on both synthetic and real datasets.

**Weaknesses**

- CARE Assumption: Realism of the CARE condition is questionable.

- Noise Assumptions: Results rely on Gaussian noise; unclear how well it generalizes for what concerns theory.

Overall, I think the strengths outweigh the weaknesses, and thus I am inclined toward acceptance.